# Biomedical Evaluation of Early Chronic Kidney Disease in the Air Force: Building a Predictive Model from the Taiwan Military Health Service

**DOI:** 10.3390/bioengineering11030231

**Published:** 2024-02-28

**Authors:** Po-Jen Hsiao, Ruei-Lin Wang, Fu-Kang Hu, Fu-Ru Tsai, Chih-Chien Chiu, Wen-Fang Chiang, Kun-Lin Wu, Yuan-Kuei Li, Jenq-Shyong Chan, Chi-Ming Chu, Chi-Wen Chang

**Affiliations:** 1Division of Nephrology, Department of Medicine, Armed Forces Taoyuan General Hospital, Taoyuan 325, Taiwan; doc10510@aftygh.gov.tw (P.-J.H.); wfc96076@yahoo.com.tw (W.-F.C.); ndmc6217316@yahoo.com.tw (K.-L.W.); jschan0908@yahoo.com.tw (J.-S.C.); 2Division of Nephrology, Department of Medicine, Tri-Service General Hospital, National Defense Medical Center, Taipei 114, Taiwan; 3Institute of Cellular and System Medicine, National Health Research Institutes, Miaoli County 350, Taiwan; 4Department of Life Sciences, National Central University, Taoyuan 320, Taiwan; 5Department of Medicine, Armed Forces Taoyuan General Hospital, Taoyuan 325, Taiwan; aassukw@livemail.tw; 6School of Public Health, National Defense Medical Center, Taipei 114, Taiwan; u272901@gmail.com (F.-K.H.); chuchiming@web.de (C.-M.C.); 7Department of Nursing, Armed Forces Taoyuan General Hospital, Taoyuan 325, Taiwan; smilefuru1010@gmail.com; 8School of Nursing, College of Medicine, Chang Gung University, Taoyuan 333, Taiwan; 9Division of Infectious Disease, Department of Internal Medicine, Taoyuan Armed Forces General Hospital, Taoyuan 325, Taiwan; calebchiu.tw@gmail.com; 10Division of Infectious Disease, Department of Internal Medicine, Tri-Service General Hospital, National Defense Medical Center, Taipei 114, Taiwan; 11Division of Colorectal Surgery, Department of Surgery, Taoyuan Armed Forces General Hospital, Taoyuan 325, Taiwan; nkcell1997@gmail.com; 12Department of Biomedical Sciences and Engineering, National Central University, Taoyuan 320, Taiwan; 13Graduate Institute of Life Sciences, National Defense Medical Center, Taipei 114, Taiwan; 14Graduate Institute of Medical Sciences, National Defense Medical Center, Taipei 114, Taiwan; 15Department of Public Health, School of Public Health, China Medical University, Taichung 404, Taiwan; 16Department of Public Health, Kaohsiung Medical University, Kaohsiung 807, Taiwan; 17Big Data Research Center, Fu-Jen Catholic University, New Taipei City 242, Taiwan; 18Division of Biostatistics and Medical Informatics, Department of Epidemiology, National Defense Medical Center, Taipei 114, Taiwan; 19Department of Healthcare Administration and Medical Informatics, Kaohsiung Medical University, 100, Shi-Chuan 1st Road, Kaohsiung 807, Taiwan; 20Division of Pediatric Endocrinology & Genetics, Department of Pediatrics, Chang-Gung Memorial Hospital, Taoyuan 333, Taiwan

**Keywords:** Air Force, chronic kidney disease, predictive model, military health

## Abstract

Objective: Chronic kidney disease (CKD) is one of the most common diseases worldwide. The increasing prevalence and incidence of CKD have contributed to the critical problem of high medical costs. Due to stressful environments, aircrew members may have a high risk of renal dysfunction. A better strategy to prevent CKD progression in Air Force personnel would be to diagnosis CKD at an early stage. Since few studies have been conducted in Taiwan to examine the long-term trends in early CKD in Air Force aircrew members, this study is highly important. We investigated the prevalence of CKD and established a predictive model of disease variation among aircrew members. Materials and Methods: In this retrospective study, we included all subjects who had received physical examinations at a military hospital from 2004 to 2010 and who could be tracked for four years. The Abbreviated Modification of Diet in Renal Disease Formula (aMDRD) was used to estimate the glomerular filtration rate (GFR) and was combined with the National Kidney Foundation/ Kidney Disease Outcomes Quality Initiative (NKF-K/DOQI) to identify CKD patients. Results: A total of 212 aircrew members were assessed. The results showed that the prevalence of CKD was 3.8%, 9.4%, 9.0%, and 9.4% in each of the four years. According to the logistic regression analysis, abnormal urobilinogen levels, ketones, and white blood cell (WBC) counts in urine and a positive urine occult blood test increased the risk of CKD. A positive urine occult blood test can be used to predict the future risk of CKD. Moreover, the generalized estimating equation (GEE) model showed that a greater risk of CKD with increased examination time, age and seniority had a negative effect. In conclusion, abnormal urobilinogen levels, ketones, and urine WBC counts in urine as well as a positive urine occult blood test might serve as independent predictors for CKD. Conclusion: In the future, we can focus not only on annual physical examinations but also on simple and accurate examinations, such as urine occult blood testing, to determine the risk of CKD and prevent its progression in our aircrew members.

## 1. Introduction

Due to changes in the lifestyle and eating habits of Taiwanese people, the occurrence of flight safety accidents caused by chronic kidney disease (CKD) and related chronic diseases among aircrew members has become a major concern in Taiwan [1]; therefore, managing the health status of aircrew members is imperative for maintaining the strength of the National Army. The Ministry of National Defense requires all aircrew members to complete an annual aircrew physical examination to ensure flight safety. Since the Air Force aircrew shoulders the primary task of the first battle in modern warfare, they must be strictly screened and are generally healthier than the general public, but do aircrew members have any increased health risk or increased risk of illness from prolonged exposure to the flight environment?

This study aimed to investigate the prevalence of CKD and its risk factors by studying the long-term trends in relevant physical examination indicators revealed by the annual physical examinations of Air Force aircrew members to provide a reference for the health management of Taiwanese military units and individuals. This study also established a prediction model for early detection and early treatment to maintain the health of aircrew members and reduce flight safety accidents. The objectives of this study were as follows:Understanding the demographic characteristics of Air Force aircrew members and the distribution of physical examination indicators.Understanding the correlations among demographic characteristics, physical examination indicators, and CKD among Air Force aircrew members.Exploring the risk factors for CKD in Air Force aircrew members and long-term trends.Establishing a long-term trend prediction model for CKD.

## 2. Study Design and Methods

This study was a period prevalence study. The research subject was defined, after which the research structure, research hypothesis, and operational definitions of the research variables were described according to the literature and data. The data collection, data processing, and data analysis steps were described. The contents of this paper are as follows:

### 2.1. Research Subjects

This study included all aircrew members of the Air Force as the population, and collected physical examination data for aircrew personnel who underwent physical examinations at four Taiwanese military hospitals (including branches) between 2004 and 2010, i.e., Taoyuan Armed Forces General Hospital, Taichung Armed Forces General Hospital, Kaohsiung Armed Forces General Hospital, and Hualien Armed Forces General Hospital.

### 2.2. Research Architecture

After a literature review, this study aimed to explore the correlations between CKD and demographic characteristics, physical examination indicators, and predictors. The research hypotheses were formulated with reference to this research framework.

### 2.3. Data Collection

The test data were collected by researchers from the aviation and medical departments of various military hospitals without direct contact with the research subjects, and the results are presented in groups without involving personal privacy. In addition, this study was approved by the Human Experimentation Committees of the Tri-Service General Hospital, to safeguard the relevant rights and interests of the research subjects.

The study subjects’ data included the following: basic personal information (including date of birth, sex, unit location, and annual physical examination date; the name and identity card number were replaced by serial numbers) and annual aircrew physical examination indicators (including height, weight, blood pressure (BP), physical examination, and laboratory tests) to facilitate the confirmation of CKD. To avoid overdiagnosis, CKD was defined as the presence of kidney damage based on an estimated glomerular filtration rate (eGFR) of less than 60 ml/min/1.73 m^2^, or evident proteinuria/haematuria persisting for more than 3 months. In this study, the eGFR was calculated according to the Abbreviated Modification of Diet in Renal Disease Formula (aMDRD): eGFR = 186 × Creatinine^−1.154^ × Age^−0.203^ (male) [2].

Electronic information (archives) on the research subjects was stored on the project’s hard drive and was maintained, analysed, and processed by dedicated graduate students. Finally, at the end of this study, the files for the research subjects were encrypted and stored, and the original files were deleted after the study results were published, which is in line with the principles of ethical confidentiality and privacy.

### 2.4. Data Processing and Analysis

First, the physical examination data were archived using EXCEL 2010 and were then statistically analysed using IBM SPSS 21.0 statistical software. The significance level was set at 0.05, and two-tailed tests were used. The following data analysis and statistical methods were employed depending on the research purpose:

#### 2.4.1. Descriptive Statistics

Categorical variables: the demographic characteristics of the research subjects (such as age, aircraft type, and unit location) and abnormal physical examination indicators are presented as numbers and percentages (%).Continuous variables, such as age and physical examination indicators (physical, urine, and blood biochemical tests), are presented as the mean ± standard deviation (mean ± SD).

#### 2.4.2. Inferential Statistics

T test, chi-square test, or analysis of variance (ANOVA): statistical analysis was performed for demographic characteristics and CKD according to the variable attributes, after which the distributions were compared.Linear regression: the relationships between physical examination indicators and serum creatinine (SCr) concentration, age group, and physical examination time (sequence) were analysed.Spearman correlation coefficient: the relationships between risk factors and renal function were analysed.Logistic regression: the risk factors for CKD at the year of physical examination and at one year, two years, and three years after physical examination were explored.Generalized estimating equation: a long-term trend prediction model for CKD was established.

## 3. Study Results

### 3.1. Description of Basic Data

After excluding duplicate data and incomplete data (including age, SCr, urine protein, urine red blood cells (RBCs), and urine pH), 212 aircrew members (all male) with four-year physical examination records were included, and the results of the fourth (most recent) physical examination were analysed (Figure 1). 

#### 3.1.1. Demographic Characteristics

The demographic characteristics of the study subjects are shown in Appendix A. The overall mean age was 40.44±7.15 years, with 100 members (47.2%) aged 30-39 years, 82 members (38.7%) aged 40-49 years, and 30 members (14.2%) aged over 50 years. The cohort included 50 members (23.6%) from Taoyuan Armed Forces General Hospital (Hsinchu Branch), 23 (10.8%) from Taichung Armed Forces General Hospital, 126 (59.4%) from Kaohsiung Armed Forces General Hospital (Gangshan Branch), and 13 (6.1%) from Hualien Armed Forces General Hospital. In terms of unit location, 50 (23.6%) were in the northern region, 23 (10.8%) were in the central region, 126 (59.4%) were in the southern region, and 13 (6.1%) were in the Huadong region. Moreover, 121 (57.1%) individuals were from fighter units, and 91 (42.9%) were from nonfighter units.

#### 3.1.2. Physical Examination

The mean height was 173.23 ± 5.52 cm. The mean body weight was 75.30 ± 8.47 kg, and the mean body mass index (BMI) was 25.06 ± 2.17 kg/m^2^. The mean waist circumference was 80.52 ± 5.47 cm. The mean systolic BP (SBP) was 127.51 ± 7.60 mmHg, and the mean diastolic BP (DBP) was 79.67 ± 6.31 mmHg. Abnormal BP (hypertension) was defined as an SBP of 130 mmHg or a DBP of 85 mmHg; therefore, 27 people (49.1%) had normal BP, and 28 (50.9%) had abnormal BP.

#### 3.1.3. Urinalysis Tests 

The results of the study subjects’ routine urine tests are shown in Appendix A. Overall, 197 members (92.9%) had normal urine protein levels, and 15 (7.1%) had abnormal urine protein levels. All members had normal urine glucose levels. Moreover, 194 members (91.5%) had normal urobilinogen levels, and 18 (8.5%) had abnormal urobilinogen levels. Overall, 201 members (94.8%) had normal levels of bilirubin in their urine, whereas 11 (5.2%) had abnormal levels of bilirubin in their urine. Ketones in the urine were normal in 194 (91.5%) individuals and abnormal in 18 (8.5%) individuals. A normal urine pH was detected in 211 patients (99.5%), while an abnormal urine pH was detected in one (0.5%) individual. Furthermore, 208 members (98.1%) had a normal urine specific gravity, and four (1.9%) had an abnormal urine specific gravity. Urine occult blood test results were normal in 186 (87.7%) individuals and abnormal in 26 (12.3%) individuals. Urine white blood cell (WBC) counts were normal in 201 (94.8%) patients and abnormal in 11 (5.2%) patients. Furthermore, 210 members (99.1%) had normal urine RBC counts, and two (0.9%) had abnormal urine RBC counts.

#### 3.1.4. Complete Blood Count Tests

The complete blood count test results of the subjects are shown in Appendix A. The mean WBC count was 6.66 ± 1.62 × 10^3^/µL, and 203 (95.8%) aircrew members had a normal WBC count, while 9 (4.2%) had an abnormal WBC count. The mean RBC count was 5.19 ± 0.45 × 10^6^/µL, and 187 (88.2%) had a normal RBC count, while 25 (11.8%) had an abnormal RBC count. The mean haemoglobin level was 15.66 ± 0.96 g/dL, and 205 (96.7%) had a normal haemoglobin level, while 7 (3.3%) had an abnormal haemoglobin level. The mean haematocrit (Hct) was 45.32 ± 2.48%, and 205 (96.7%) had normal Hct, while 7 (3.3%) had abnormal Hct. The mean corpuscular haemoglobin (MCH) was 30.39 ± 2.59 pg, and 153 (93.9%) had normal MCH, while 10 (6.1%) had abnormal MCH. The mean corpuscular volume (MCV) was 87.69 ± 6.68 fl, and 198 (93.4%) had a normal MCV, while 14 (6.6%) had an abnormal MCV. The mean corpuscular haemoglobin concentration (MCHC) was 34.50 ± 0.93 g/dL, and 160 (98.8%) had a normal MCHC, while 2 (1.2%) had an abnormal MCHC. The mean platelet count (PLT) was 237.44 ± 51.30 × 10^3^/µL, and 205 (96.7%) had a normal PLT, while 7 (3.3%) had an abnormal PLT.

#### 3.1.5. Blood Biochemical Tests

The blood biochemical test results of the subjects are shown in Appendix A. The mean fasting blood glucose (FBG) was 97.08 ± 9.67 mg/dL, and 138 (65.1%) aircrew members had normal FBG, while 74 (34.9%) had abnormal FBG. The mean blood urea nitrogen (BUN) level was 13.73 ± 2.95 mg/dL, and 208 (98.1%) had normal BUN, while 4 (1.9%) had abnormal BUN. The mean SCr was 0.98 ± 0.14 mg/dL, and 191 (90.1%) had normal SCr, but 21 (9.9%) had abnormal SCr. The mean uric acid level was 6.10 ± 1.18 mg/dL, and 169 (79.7%) had normal uric acid, while 43 (20.3%) had abnormal uric acid. The mean aspartate aminotransferase/serum glutamic oxaloacetic transaminase (AST/SGOT) was 25.18 ± 10.07 U/L, and 199 (94.3%) had normal AST, while 12 (5.7%) had abnormal AST. The mean alanine aminotransferase/serum glutamate pyruvate transaminase (ALT/SGPT) level was 32.94 ± 25.60 U/L, and 168 (79.6%) had normal ALT, while 43 (20.4%) had abnormal ALT. The mean alkaline phosphatase (ALP) level was 65.11 ± 16.66 U/L, and all 212 had normal ALP. The mean total protein (TP) level was 7.38 ± 0.40 g/dL, and 209 (98.6%) had normal TP, while 3 (1.4%) had abnormal TP. The mean albumin (Alb) level was 4.70 ± 0.25 g/dL, and while 183 (86.3%) had normal Alb, 29 (13.7%) had abnormal Alb. The mean total bilirubin (TBIL) level was 0.76 ± 0.37 mg/dL, and while 185 (87.3%) had normal TBIL, 27 (12.7%) had abnormal TBIL. The mean triglyceride (TG) level was 122.50 ± 66.54 mg/dL; 155 (73.1%) had normal TGs, and 57 (26.9%) had abnormal TGs. The mean total cholesterol (TCH) level was 191.29 ± 27.57 mg/dL; 136 (64.2%) had normal TCH, and 76 (35.8%) had abnormal TCH. The mean high-density lipoprotein cholesterol (HDLC) level was 50.27 ± 13.00 mg/dL; 164 (77.4%) had normal HDLC, and 48 (22.6%) had abnormal HDLC. The mean low-density lipoprotein cholesterol (LDLC) level was 125.62 ± 26.27 mg/dL; 122 (57.5%) had normal LDLC, and 90 (42.5%) had abnormal LDLC.

### 3.2. Correlations among SCr Concentration, Physical Examination Indicators, and CKD

The results of the basic examination, routine urine test, and blood biochemical test were analysed according to the SCr quartiles (the first quartile is <0.90 mg/dL, the second quartile is ≥0.90 and <1.00 mg/dL, the third quartile is ≥1.00 and <1.06 mg/dL, and the fourth quartile is ≥1.06 mg/dL). A total of 47 aircrew members were grouped into the first quartile. The mean age was 40.60 ± 6.95 years: 22 aircrew members (46.8%) were in the 30–39-year age group, 18 (38.3%) were in the 40–49-year age group, and 7 (14.9%) were over 50 years of age. The mean BMI was 25.15 ± 2.22 kg/m^2^. The mean SBP was 126.81 ± 6.15 mmHg: nine aircrew members (56.3%) had normal SBP and seven (43.8%) had abnormal SBP. The mean DBP was 79.50 ± 5.44 mmHg: 12 aircrew members (75.0%) had normal DBP and 4 (25.0%) had abnormal DBP. Overall, nine aircrew members (56.3%) had normal BP and seven (43.8%) had hypertension. Four aircrew members (8.5%) were diagnosed with CKD. Fifty-five aircrew members were grouped into the second quartile. The mean age was 38.76 ± 5.74 years: 30 aircrew members (54.5%) were in the 30–39-year age group, 22 (40.0%) were in the 40–49-year age group, and 3 were over 50 years of age (5.5%). The mean BMI was 25.03 ± 2.40 kg/m^2^. The mean SBP was 126.04 ± 8.37 mmHg: 13 aircrew members (59.1%) had normal SBP and 9 (40.9%) had abnormal SBP. The mean DBP was 79.04 ± 6.60 mmHg: 17 aircrew members (77.3%) had normal DBP and 5 (22.7%) had abnormal DBP. Overall, 12 aircrew members (54.5%) had normal BP and 10 (45.5%) had hypertension. Eight aircrew members (14.5%) were diagnosed with CKD. Fifty-five aircrew members were grouped into the third quartile. The mean age was 40.85 ± 7.89 years: 27 aircrew members (49.1%) were in the 30–39-year age group, 17 (30.9%) were in the 40–49-year age group, and 11 were over 50 years of age (20.0%). The mean BMI was 25.33 ± 1.05 kg/m^2^. The mean SBP was 131.00 ± 6.54 mmHg: one aircrew member (16.7%) had normal SBP and five (83.3%) had abnormal SBP. The mean DBP was 80.50 ± 7.53 mmHg: three aircrew members (50.0%) had normal DBP and three (50.0%) had abnormal DBP. Overall, one aircrew member (16.7%) had normal BP and five (83.3%) had hypertension. Three aircrew members (5.5%) were diagnosed with CKD. Fifty-five aircrew members were grouped into the fourth quartile. The mean age was 41.56 ± 7.70 years: 21 aircrew members (38.2%) were in the 30–39-year age group, 25 (45.5%) were in the 40–49-year age group, and 9 (16.4%) were over 50 years of age. The mean BMI was 24.86 ± 2.25 kg/m^2^. The mean SBP was 129.54 ± 8.30 mmHg: five aircrew members (45.5%) had normal SBP and six (64.5%) had abnormal SBP. The mean DBP was 80.72 ± 6.97 mmHg: eight aircrew members (72.7%) had normal DBP and three (27.3%) had abnormal DBP. Overall, five aircrew members (45.5%) had normal BP and six (54.5%) had hypertension. Five aircrew members (9.1%) were diagnosed with CKD (Appendix A).

The test results of aircrew members in the first quartile are as follows: four (8.5%) had abnormal urine protein, one (2.1%) had abnormal urobilinogen, one (2.1%) had abnormal ketones in the urine, one (2.1%) had abnormal urine WBCs, and one (2.1%) had abnormal urine RBCs; the mean urine pH was 5.93 ± 0.77, and no aircrew members had an abnormal urine pH; 10 (21.3%) had urine occult blood. 

The test results of aircrew members in the second quartile are as follows: six (10.9%) had abnormal urine protein, eight (14.5%) had abnormal urobilinogen, two (3.6%) had abnormal bilirubin, and two (3.6%) had abnormal ketones in the urine; the mean urine pH was 6.09 ± 0.80, and one (1.8%) aircrew member had an abnormal urine pH; four (7.3%) had abnormal urine WBCs, 12 (21.8%) had urine occult blood, and one (1.8%) had abnormal urine RBCs. 

The test results of aircrew members in the third quartile are as follows: three (5.5%) had abnormal urine protein, three (5.5%) had abnormal bilirubin in the urine, five (9.1%) had abnormal urobilinogen, and seven (12.7%) had abnormal ketones in the urine; the mean urine pH was 5.71 ± 0.66, and no aircrew members had an abnormal urine pH; two (3.6%) had abnormal urine WBCs, and urine occult blood was detected in two (3.6%). 

The test results of aircrew members in the fourth quartile are as follows: two (3.6%) had abnormal urine protein, two (3.6%) had urine occult blood, four (7.3%) had abnormal urobilinogen, four (7.3%) had abnormal urine WBCs, six (10.9%) had abnormal bilirubin in the urine, and eight (14.5%) had abnormal ketones in the urine; the mean urine pH was 5.72 ± 0.77, and no aircrew members had abnormal urine pH. Routine urine test data showed that the mean urine pH was significantly different for different SCr quartiles (*p* = 0.026) and decreased as the SCr concentration increased (*p* = 0.029). Abnormal ketones in the urine and urine occult blood detection were significantly different among SCr quartiles, with *p* values of 0.046 and 0.001, respectively (Appendix A). The test results of aircrew members in the first quartile are as follows: the mean FBG was 93.55 ± 12.01 mg/dL, and 14 (29.8%) had hyperglycaemia; the mean BUN level was 12.92 ± 2.38 mg/dL, and no aircrew members had abnormal BUN; the mean uric acid level was 5.09 ± 1.16 mg/dL, and 7 (14.9%) had abnormal uric acid; the mean ALT (SGPT) level was 39.78 ± 37.60 U/L, and 12 (25.5%) had abnormal ALT; the mean TBIL level was 0.71 ± 0.26 g/dL, and 4 (8.5%) had abnormal TBIL; the mean TG level was 122.42 ± 59.10 mg/dL, and 13 (27.7%) had abnormal TG; the mean TCH level was 190.73 ± 28.90 mg/dL, and 15 (31.9%) had abnormal TCH; the mean HDLC level was 50.67 ± 12.84 mg/dL, and 11 (23.4%) had abnormal HDLC; the mean LDLC level was 130.21 ± 27.34 mg/dL, and 22 (46.8%) had abnormal LDLC (Appendix A). 

The test results of aircrew members in the second quartile are as follows: the mean FBG was 97.77 ± 9.46 mg/dL, and 17 (30.9%) had hyperglycaemia; the mean BUN level was 13.05 ± 3.38 mg/dL, and 1 (1.8%) had abnormal BUN; the mean uric acid level was 5.92 ± 1.15 mg/dL, and 10 (18.2%) had abnormal uric acid; the mean ALT (SGPT) level was 32.90 ± 24.24 U/L, and 12 (22.2%) had abnormal ALT; the mean TBIL level was 0.77 ± 0.40 g/dL, and 6 (10.9%) had abnormal TBIL; the mean TG level was 129.01 ± 67.12 mg/dL, and 17 (30.9%) had abnormal TG; the mean TCH level was 193.15 ± 24.56 mg/dL, and 18 (32.7%) had abnormal TCH; the mean HDLC level was 49.10 ± 13.24 mg/dL, and 11 (20.0%) had abnormal HDLC; the mean LDLC level was 130.44 ± 25.15 mg/dL, and 26 (47.3%) had abnormal LDLC. 

The test results of aircrew members in the third quartile are as follows: the mean FBG was 99.53 ± 8.60 mg/dL, and 26 (47.3%) had hyperglycaemia; the mean BUN level was 14.15 ± 2.77 mg/dL, and 1 (1.8%) had abnormal BUN; the mean uric acid level was 5.87 ± 1.02 mg/dL, and 8 (14.5%) had abnormal uric acid; the mean ALT (SGPT) level was 32.19 ± 19.25 U/L, and 12 (21.8%) had abnormal ALT; the mean TBIL level was 0.82 ± 0.45 g/dL, and 9 (16.4%) had abnormal TBIL; the mean TG level was 110.73 ± 46.13 mg/dL, and 13 (23.6%) had abnormal TG; the mean TCH level was 190.52 ± 30.52 mg/dL, and 22 (40.0%) had abnormal TCH; the mean HDLC level was 49.98 ± 12.99 mg/dL, and 15 (23.7%) had abnormal HDLC; the mean LDLC level was 123.93 ± 25.93 mg/dL, and 24 (43.6%) had abnormal LDLC. 

The test results of aircrew members in the fourth quartile are as follows: the mean FBG was 96.94 ± 7.84 mg/dL, and 17 (30.9%) had hyperglycaemia; the mean BUN level was 14.68 ± 2.84 mg/dL, and 2 (3.6%) had abnormal BUN; the mean uric acid level was 6.69 ± 1.20 mg/dL, and 18 (32.7%) had abnormal uric acid; the mean ALT (SGPT) level was 27.89 ± 18.17 U/L, and 7 (12.7%) had abnormal ALT; the mean TBIL level was 0.73 ± 0.35 g/dL, and 8 (14.5%) had abnormal TBIL; the mean TG level was 127.84 ± 86.47 mg/dL, and 14 (25.5%) had abnormal TG; the mean TCH level was 190.48 ± 26.82 mg/dL, and 21 (38.2%) had abnormal TCH; the mean HDLC level was 51.39 ± 13.14 mg/dL, and 11 (20.0%) had abnormal HDLC; the mean LDLC level was 118.61 ± 25.69 mg/dL, and 18 (32.7%) had abnormal LDLC. Blood biochemical test data showed that the mean FBG, BUN, and uric acid levels were significantly different for different SCr quartiles, with *p* values of 0.017, 0.004, and <0.001, respectively. In addition, the mean BUN and uric acid levels tended to increase as the SCr quartile increased, with *p* values of <0.001 and 0.001, respectively, while the mean ALT (SGPT) and LDLC levels tended to decrease as the SCr quartile increased, with *p* values of 0.025 and 0.010, respectively. 

### 3.3. Correlations among Age, Physical Examination Indicators, and CKD

Appendix A shows the correlation between age and CKD; the results of the basic examination, routine urine test, and blood biochemical test were analysed according to the three age groups (30–39 years, 40–49 years, and over 50 years). In addition, the BMI and BP in the basic examination of aircrew members over the age of 50 were missing, and thus, only the relationship between age and CKD of these aircrew members was analysed. In the 30–39-year age group, the mean glomerular filtration rate (GFR) was 96.29 ± 14.53 mL/min/1.73 m^2^: 73 (73.0%) had a GFR ≥ 90 mL/min/1.73 m^2^, 26 (26.0%) had a GFR of 60–89 mL/min/1.73 m^2^, and 1 (1.0%) had a GFR of 45–59 mL/min/1.73 m^2^. Twelve (12.0%) were diagnosed with CKD, with eight (8.0%) in stage 1, three (3.0%) in stage 2, and one (1.0%) in stage 3. In the 40–49-year age group, the mean GFR was 90.43 ± 15.89 mL/min/1.73 m^2^: 36 (43.9%) had a GFR ≥ 90 mL/min/1.73 m^2^, 44 (53.7%) had a GFR of 60–89 mL/min/1.73 m^2^, and 2 (2.4%) had a GFR of 45–59 mL/min/1.73 m^2^. Eight (9.8%) were diagnosed with CKD, with two (2.4%) in stage 1, four (4.9%) in stage 2, and two (2.4%) in stage 3. In the group over 50 years of age, the mean GFR was 90.05 ± 19.34 mL/min/1.73 m^2^: 10 (33.3%) had GFR ≥ 90 mL/min/1.73 m^2^, and 20 (66.7%) had a GFR of 60–89 mL/min/1.73 m^2^; no aircrew members were diagnosed with CKD. The basic examination data showed that the mean GFR was significantly different among age groups (*p* = 0.025) and that the mean GFR declined with increasing age (*p* = 0.013). In addition, in each age group, the groups based on GFR were significantly different (*p* < 0.001).

Appendix A shows the correlation between the age and routine urine tests; in the 30–39-year age group, ten (10.0%) had abnormal urine protein, ten (10.0%) had abnormal urobilinogen, five (5.0%) had abnormal bilirubin in the urine, and six (6.0%) had abnormal ketones in the urine; the mean urine pH was 5.93 ± 0.69, and no aircrew members had abnormal urine pH; five (5.0%) had abnormal urine WBCs, 17 (17.0%) had urine occult blood, and two (2.0%) had abnormal urine RBCs. In the 40–49-year age group, five (6.1%) had abnormal urine protein, seven (8.5%) had abnormal urobilinogen, four (4.9%) had abnormal bilirubin in the urine, and nine (11.0%) had abnormal ketones in the urine; the mean urine pH was 5.86 ± 0.86, and one (1.2%) had abnormal urine pH; six (7.3%) had abnormal urine WBCs; and eight (9.8%) had urine occult blood. In the group over 50 years of age, all aircrew members had normal urine protein, urine WBCs, and urine RBCs; one (3.3%) had abnormal urobilinogen, one (3.3%) had urine occult blood, two (6.7%) had abnormal bilirubin in the urine, and three (10.0%) had abnormal ketones in the urine; the mean urine pH was 5.61 ± 0.69, and no aircrew members had abnormal urine pH. Routine urine test data showed that the mean urine pH decreased with increasing age, but the trend was not significantly different. The abnormalities and mean values of other items in the routine urine test were not significantly different among different age groups. 

Appendix A shows the correlation between age and blood biochemical tests. In the 30–39-year age group, the mean FBG level was 95.40 ± 8.46 mg/dL, and 27 (27.0%) had hyperglycaemia; the mean BUN level was 13.72 ± 3.12 mg/dL, and 3 (3.0%) had abnormal BUN; the mean SCr level was 0.97 ± 0.13 mg/dL, and 8 (8.0%) had abnormal SCr; the mean uric acid level was 6.19 ± 1.08 mg/dL, and 24 (24.0%) had abnormal uric acid; the mean ALT (SGPT) level was 34.66 ± 29.64 U/L, and 22 (22.2%) had abnormal ALT; the mean TBIL level was 0.75 ± 0.37 g/dL, and 8 (8.0%) had an abnormal TBIL level; the mean TG level was 113.54 ± 47.99 mg/dL, and 24 (24.0%) had abnormal TG; the mean TCH level was 187.06 ± 27.28 mg/dL, and 26 (26.0%) had abnormal TCH; the mean HDLC level was 50.51 ± 11.63 mg/dL, and 20 (20.0%) had abnormal HDLC; the mean LDLC level was 123.14 ± 26.15 mg/dL, and 35 (35.0%) had abnormal LDLC. 

In the 40–49-year age group, the mean FBG level was 97.77 ± 10.92 mg/dL, and 30 (36.6%) had hyperglycaemia; the mean BUN level was 13.60 ± 2.62 mg/dL, and no aircrew members had abnormal BUN; the mean SCr level was 0.99 ± 0.15 mg/dL, and 11 (13.4%) had abnormal SCR; the mean uric acid level was 6.08 ± 1.31 mg/dL, and 14 (17.1%) had abnormal uric acid; the mean ALT (SGPT) level was 30.54 ± 18.88 U/L, and 15 (18.3%) had abnormal ALT; the mean TBIL level was 0.79 ± 0.38 g/dL, and 15 (18.3%) had abnormal TBIL; the mean TG level was 133.67 ± 86.45 mg/dL, and 26 (31.7%) had abnormal TG; the mean TCH level was 194.51 ± 24.56 mg/dL, and 37 (45.1%) had abnormal TCH; the mean HDLC level was 50.86 ± 14.91 mg/dL, and 21 (25.6%) had abnormal HDLC; the mean LDLC level was 127.12 ± 24.31 mg/dL, and 41 (50.0%) had abnormal LDLC. 

In the group over 50 years of age, the mean FBG level was 100.75 ± 8.82 mg/dL, and 17 (56.7%) had hyperglycaemia; the mean BUN level was 14.06 ± 3.28 mg/dL, and 1 (3.3%) had abnormal BUN; the mean SCr level was 0.96 ± 0.15 mg/dL, and 2 (6.7%) had abnormal SCr; the mean uric acid level was 5.86 ± 1.09 mg/dL, and 5 (16.7%) had abnormal uric acid; the mean ALT (SGPT) level was 33.85 ± 27.45 U/L, and 6 (20.0%) had abnormal ALT; the mean TBIL level was 0.74 ± 0.34 g/dL, and 4 (13.3%) had abnormal TBIL; the mean TG level was 121.88 ± 53.36 mg/dL, and 7 (23.3%) had abnormal TG; the mean TCH level was 196.61 ± 34.43 mg/dL, and 13 (43.3%) had abnormal TCH; the mean HDLC level was 47.86 ± 11.81 mg/dL, and 7 (23.3%) had abnormal HDLC; the mean LDLC level was 129.75 ± 31.49 mg/dL, and 14 (46.7%) had abnormal LDLC. Blood biochemical test data showed that the mean FBG level was significantly different among the various age groups (*p* = 0.020) and increased with increasing age (*p* = 0.005). The mean TCH level also tended to increase with increasing age (*p* = 0.041). In addition, the abnormalities in FBG and TCH were significantly different among the age groups. 

This section presents the analyses of the consecutive four-year physical examination results of 212 aircrew members and compares the basic examination, routine urine test (Appendix A), and blood biochemical test results (Appendix A). Among the study subjects, the mean BMI was highest during the second physical examination (25.47 ± 2.34 kg/m^2^) and was lowest during the fourth physical examination (25.06 ± 2.17 kg/m^2^); the mean SBP was highest at the first physical examination (129.30 ± 9.45 mmHg) and was lowest at the third physical examination (127.18 ± 9.60 mmHg); the mean DBP was highest at the first physical examination (80.75 ± 6.70 mmHg) and lowest at the third physical examination (77.49 ± 8.28 mmHg). Among those with hypertension, the highest proportion (34, 60.7%) was observed during the first physical examination, while the lowest proportion (25, 45.5%) was observed during the third physical examination. Among the study subjects, the proportion of subjects with abnormal urine protein was highest at the fourth physical examination (15, 7.1%) and lowest at the first physical examination (8, 3.8%); the proportion of subjects with abnormal urobilinogen was highest at the fourth physical examination (18, 8.5%) and lowest at the first physical examination (9, 4.2%); the proportion of subjects with abnormal bilirubin in the urine was highest at the third physical examination (13, 6.1%) and lowest at the second physical examination (9, 4.2%); the proportion of those with abnormal ketones in the urine was highest in the second physical examination (19, 9.0%) and lowest at the first physical examination (10, 4.7%); the mean urine pH was highest at the fourth physical examination (5.86 ± 0.76 mg/dL) and lowest at the first physical examination (5.81 ± 0.79 mg/dL); the proportion of those with abnormal urine WBCs was highest at the third physical examination (30, 14.2%); the proportion of aircrew members with urine occult blood detected was highest during the third physical examination (29, 13.7%) and lowest during the first physical examination (18, 8.5%); the proportion of aircrew members with abnormal urine RBCs was highest in the second physical examination (8, 3.8%)

Among the study subjects, both the mean FBG level of 97.08 ± 9.67 mg/dL and the proportion of aircrew members (74, 34.9%) with abnormal FBG were highest at the fourth physical examination; both the mean BUN level of 14.08 ± 3.07 mg/dL and the proportion of abnormal cases (12, 5.7%) were highest at the first physical examination; both the mean SCr level of 0.98 ± 0.14 mg/dL and the proportion of abnormal cases (21, 9.9%) were highest at the fourth physical examination; both the mean uric acid level of 6.47 ± 1.29 mg/dL and the proportion of abnormal cases (70, 33.0%) were highest at the first physical examination; both the mean ALT (SGPT) level of 32.94 ± 25.60 U/L and the proportion of abnormal cases (43, 20.4%) were highest in the fourth physical examination; the mean TBIL level of 0.88 ± 0.37 g/dL was highest in the first physical examination, and the proportion of abnormal cases (16, 20.5%) was highest in the second physical examination; the mean TG level of 121.46 ± 66.24 mg/dL was highest at the third physical examination, and the proportion of abnormal cases (56, 26.5%) was highest in the fourth physical examination; both the mean TCH level of 194.08 ± 28.30 mg/dL and the proportion of abnormal cases (86, 40.6%) were highest in the third physical examination; the mean HDLC level of 50.27 ± 13.00 mg/dL was lowest in the fourth physical examination, and the proportion of abnormal cases (48, 22.6%) was highest in the fourth physical examination; both the mean LDLC level of 134.95 ± 24.65 mg/dL and the proportion of abnormal cases (42, 55.3%) were highest in the first physical examination. The results revealed that differences in the abnormalities in urine WBCs, urine RBCs, FBG, SCr, and uric acid in previous physical examinations (along the time sequence) were significant. The mean FBG, SCr, and uric acid levels were significantly different in previous physical examinations (along the time sequence). The mean FBG, SCr, and TG levels increased (*p* < 0.001) along the time sequence of physical examinations (*p* < 0.001), while the mean uric acid and TBIL levels decreased (*p* < 0.05). 

The comparison of the CKD correlation in previous physical examinations is shown in Table 1. The mean GFR of 93.14 ± 16.01 mL/min/1.73 m^2^ in the fourth physical examination was the lowest and that of 107.29 ± 19.34 mL/min/1.73 m^2^ in the first physical examination was the highest. The prevalence of CKD was 3.8% (8) for the first physical examination, 9.4% (20) for the second physical examination, 9.0% (19) for the third physical examination, and 9.4% (20) for the fourth physical examination. In terms of the CKD stage, in the first physical examination, eight were in stage 1; in the second physical examination, fourteen were in stage 1 and six were in stage 2; in the third physical examination, eleven were in stage 1, seven were in stage 2, and one was in stage 3; in the fourth physical examination, ten were in stage 1, seven were in stage 2, and three were in stage 3. The results showed that the mean GFR was significantly different (*p* < 0.001) in all previous physical examinations, and the Scheffé test (a type of post hoc, statistical analysis test) showed that the mean GFR in the first physical examination was greater than the means found in the third and fourth physical examinations. Moreover, the mean GFR in the second physical examination was greater than that in the fourth physical examination, and the mean GFR decreased along the time sequence of physical examinations (*p* < 0.001). The mean GFR was significantly different between the CKD and non-CKD groups in all previous physical examinations (*p* < 0.001). 

### 3.4. Previous Physical Examination Indicators and the Risk of Developing CKD

Univariate analysis: The physical examination indicators with high appropriateness according to the CKD-QIC were selected by the GEE. In addition to age, nine items were included, such as abnormal urobilinogen and abnormal ketones in urine. The previous physical examination indicators were analysed by logistic regression (Table 2), and the results showed that age had a protective effect on the risk of CKD development, but the difference was not significant; compared with the aircrew with normal urobilinogen levels in each physical examination, those with abnormal urobilinogen had an increased risk of developing CKD (*p* < 0.05). In the third physical examination, the risk of CKD development in the aircrew with abnormal ketones in the urine was increased by 4.880 (*p* < 0.05); for aircrew members with abnormal urine occult blood, the risk of developing CKD was increased in the second, third, and fourth physical examinations (*p* < 0.001). In the second physical examination, the risk of CKD development in the aircrew with abnormal urine WBCs was increased 12.467-fold (*p* < 0.001). 

After controlling for age, the logistic regression showed that the risk of CKD development in aircrew members with abnormal urobilinogen was still higher than that in the aircrew with normal urobilinogen in each physical examination, and the difference was significant. Aircrew members with abnormal ketones in the urine had an increased risk of developing CKD by the second and third physical examinations (*p* < 0.05). Aircrew members with abnormal urine occult blood and abnormal urine WBCs had an increased risk of developing CKD by the second, third, and fourth physical examinations, and the differences were significant. 

Year-by-year prediction analysis: The purpose of this analysis was to determine whether the risk of CKD in the next year, two years, and three years could be predicted by abnormal (CKD) indicators at the first physical examination. Therefore, physical examination indicators in a certain year served as independent variables, and whether an individual was diagnosed with CKD in the next year, two years, or three years served as the dependent variable; moreover, a logistic regression analysis was performed (Table 3a,b). In Model 1 (using indicators in the first physical examination to predict the risk of CKD by the second physical examination), aircrew members with abnormal urobilinogen had a 5.471-fold (95% confidence interval (CI): 1.255–23.845) increased risk of developing CKD, with an explained variance (R^2^) of 0.042; aircrew members who had urine occult blood had an 8.860-fold (95% CI: 2.943–26.674) increased risk of developing CKD, with an R^2^ of 0.128; and aircrew members with CKD had a 6.600-fold (95% CI: 1.451–30.027) risk of continuing to have CKD, with an R^2^ of 0.049. After controlling for age, aircrew members with abnormal urobilinogen and urine occult blood had a 4.771-fold (95% CI: 1.068–21.317) and 7.942-fold (95% CI: 2.613–24.139) increased risk of developing CKD, with an R^2^ of 0.066 and 0.146, respectively; aircrew members with CKD had a 5.565-fold (95% CI: 1.194–25.950) increased risk of continuing to have CKD, with an R^2^ of 0.072. 

In Model 2 (using indicators in the second physical examination to predict the risk of CKD by the third physical examination), aircrew members with urine occult blood had an 8.077-fold (95% CI: 2.719–23.997) increased risk of developing CKD, with an R^2^ of 0.124; aircrew members with CKD had a 14.819-fold (95% CI: 5.022–44.152) risk of continuing to have CKD, with an R^2^ of 0.216. After controlling for age, aircrew members with urine occult blood had a 7.366-fold (95% CI: 2.398–22.625) increased risk of developing CKD, with an R^2^ of 0.127; aircrew members with CKD had a 13.958-fold (95% CI: 4.659–41.816) risk of continuing to have CKD, with an R^2^ of 0.221. 

In Model 3 (using indicators in the third physical examination to predict the risk of CKD by the fourth physical examination), aircrew members with abnormal urobilinogen had a 5.083-fold (95% CI: 1.408–18.353) increased risk of developing CKD, with an R^2^ of 0.051; aircrew members with urine occult blood had a 4.161-fold (95% CI: 1.500–11.545) increased risk of developing CKD, with an R^2^ of 0.066; high FBG had a protective effect, i.e., high FBG reduced the risk of developing CKD by 0.778-fold (95% CI: 0.050–0.987), with an R^2^ of 0.055; aircrew members with CKD had an 8.077-fold (95% CI: 2.719–23.997) risk of continuing to have CKD, with an R^2^ of 0.121. After controlling for age, aircrew members with abnormal urobilinogen and urine occult blood had a 4.618-fold (95% CI: 1.250–17.060) and a 3.875-fold (95% CI: 1.372–10.941) increased risk of developing CKD, with an R^2^ of 0.059 and 0.071, respectively; the protective effect of high FBG disappeared (became nonsignificant); aircrew members with CKD had a 7.695-fold (95% CI: 2.576–22.982) risk of continuing to have CKD, with an R^2^ of 0.127. 

In Model 4 (using indicators in the first physical examination to predict the risk of CKD by the third physical examination), aircrew members with urine occult blood had a 13.309-fold (95% CI: 4.381–40.429) increased risk of developing CKD, with an R^2^ of 0.186. After controlling for age, aircrew members with urine occult blood had a 12.374-fold (95% CI: 4.018–37.943) increased risk of developing CKD, with an R^2^ of 0.192. 

In Model 5 (using indicators in the first physical examination to predict the risk of CKD by the fourth physical examination), aircrew members with abnormal ketones in the urine had a 4.664-fold (95% CI: 1.104–19.701) increased risk of developing CKD, with an R^2^ of 0.036; aircrew members with urine occult blood had a 4.590-fold (95% CI: 1.441–14.614) increased risk of developing CKD, with an R^2^ of 0.056. After controlling for age, aircrew members with abnormal ketones in the urine and urine occult blood had a 5.004-fold (95% CI: 1.165–21.497) and a 4.252-fold (95% CI: 1.320–13.696) increased risk of developing CKD, with an R^2^ of 0.051 and 0.063, respectively. 

In Model 6, using indicators in the second physical examination to predict the risk of CKD by the fourth physical examination), aircrew members with abnormal urobilinogen had a 6.067-fold (95% CI: 1.835–20.055) increased risk of developing CKD, with an R^2^ of 0.073; aircrew with urine occult blood had a 3.933-fold (95% CI: 1.257–12.313) increased risk of developing CKD, with an R^2^ of 0.047; aircrew with abnormal urine WBCs had a 4.664-fold (95% CI: 1.104–19.701) increased risk of developing CKD, with an R^2^ of 0.036; aircrew members with abnormal BUN had a 10.556-fold (95% CI: 1.402–79.467) increased risk of developing CKD, with an R^2^ of 0.045; aircrew members with CKD had a 13.463-fold (95% CI: 4.614–39.279) risk of continuing to have CKD, with an R^2^ of 0.200. After controlling for age, aircrew members with abnormal urobilinogen, urine occult blood, abnormal urine WBCs, abnormal BUN, and CKD still had a high risk of developing CKD, which was significant.

### 3.5. Establishing a Prediction Model for the Long-Term Trends in CKD

In this section, the year-by-year prediction analysis described in the previous section continues. To establish a prediction model for the long-term trends in CKD, for the predictors of CKD (excluding age and SCr, two necessary variables for calculating GFR), the GEE was used to examine the long-term trends under repeated measurement over time, including two types of models, and the representative meanings are described below. The first type of model is the univariate analysis (Table 4a,b). Model 1 examines the physical examination time (sequence) and the long-term trends in CKD, while Models 2–9 examine each predictor over time and the long-term trends in CKD. The second type of model includes the variables with high appropriateness for CKD in the univariate analysis, such as urine occult blood, urobilinogen, urine WBCs, and ketones in the urine, to find the predictive model combination with the best QIC appropriateness. In Model 1, compared with the first physical examination, the risk of developing CKD by the second to fourth physical examinations was increased 2.656-fold (95% CI: 1.234–5.716), 2.510-fold (95% CI: 1.084–5.812), and 2.656-fold (95% CI: 1.192–5.921); the QIC was 469.00, and the differences were significant. 

For Models 2–9, each is a univariate analysis of one predictor under repeated measurement over time, and for each predictor, normal subjects are taken as the reference. The results of Models 2–5 showed that aircrew members with urine occult blood had a 6.746-fold (95% CI: 3.383–13.456) increased risk of developing CKD and that the QIC was 428.43; aircrew members with abnormal urobilinogen had a 5.614-fold (95% CI: 3.021–10.433) increased risk of developing CKD, with a QIC of 441.68; aircrew members with abnormal urine WBCs had a 3.805-fold (95% CI: 1.996–7.253) increased risk of developing CKD, with a QIC of 458.98; aircrew members with ketones in the urine had a 2.571-fold (95% CI: 1.498–5.054) increased risk of developing CKD, with a QIC of 464.62; all differences were significant. In addition, the results of Models 6–9 showed that for aircrew members with abnormal LDLC, BUN, and uric acid, the risk of developing CKD was increased, while abnormal FBG had a protective effect; however, the differences were not significant. 

According to the order of appropriateness between each variable and CKD (QIC appropriateness is high and *p* < 0.001 or *p* < 0.05), the presence of urine occult blood was first included in the combined prediction model. The results showed that combined with urine occult blood, aircrew members with abnormal urobilinogen, abnormal urine WBCs, and abnormal ketones in the urine had an increased risk of developing CKD and that QIC appropriateness was better than that of the original analysis using a single predictor (QIC was 408.94, 418.18, and 424.60, respectively, for abnormal urobilinogen, abnormal urine WBCs and abnormal ketones); moreover, the differences were significant. Combined with urine occult blood, aircrew members with abnormal LDLC, abnormal BUN, abnormal uric acid, and high FBG, high FBG, and abnormal BUN had a protective effect on CKD, but abnormal LDLC and abnormal uric acid could increase the risk of developing CKD, and the differences were statistically nonsignificant. 

Next, urine occult blood and abnormal urobilinogen were included in the combined prediction model. The results showed that combined with urine occult blood and abnormal urobilinogen, abnormal urine WBCs and abnormal ketones in the urine could increase the risk of developing CKD and that the QIC appropriateness was better than that of the previous analysis that only included urine occult blood (QIC was 399.65 and 405.22), and the differences were significant. However, combined with urine occult blood and abnormal urobilinogen, aircrew members with abnormal LDLC, abnormal BUN, abnormal uric acid, and high FBG, high FBG, and abnormal BUN had a protective effect on CKD, but abnormal LDLC and abnormal uric acid could increase the risk of developing CKD, and these differences were statistically nonsignificant. 

Then, urine occult blood, abnormal urobilinogen, abnormal urine WBCs, and abnormal ketones in the urine were included in the combined prediction model. The results showed that the QIC was 397.441. All these variables are independent predictors of CKD, and the differences were significant. In addition, repeated measurements over time found that the length of service had a protective effect on CKD, i.e., the risk of developing CKD could be reduced by 5.1% for every one-year increase in the length of service (*p* < 0.05). Finally, urine occult blood, abnormal urobilinogen, abnormal urine WBCs, abnormal ketones in the urine, and length of service were included in the combined prediction model, and the QIC was 395.418. Although the protective effect of the length of service was maintained, urine occult blood, abnormal urobilinogen, abnormal urine WBCs, and abnormal ketones in the urine could increase the risk of developing CKD, which is consistent with the above results.

### 3.6. Comparison of CKD and the Number of Abnormal Predictors

Based on the best predictors (urine occult blood, abnormal urobilinogen, abnormal urine WBCs, and abnormal ketones in the urine) found in the prediction model for the long-term trends in CKD in the previous section, the relationship between the number of abnormal predictors and CKD was analysed.

The number of aircrew members with zero abnormal predictors (Table 5a) and without CKD was 172 at the first physical examination (accounting for 92.7%), 152 at the second physical examination (accounting for 96.2%), 139 at the third physical examination (accounting for 97.2%), and 145 at the fourth physical examination (accounting for 96.0%); the negative predictive value of not having CKD was 96.65%. With one abnormal predictor, the positive predictive value of having CKD was 14.03%. With two abnormal predictors, the positive predictive value of having CKD was 41.93%, and the number of abnormal predictors was significantly different between the CKD and non-CKD groups. The logistic regression analysis (Table 5b) found that when zero as the number of abnormal predictors was used as the reference, the risk of developing CKD increased as the number of abnormal predictors increased; in the group with two abnormal predictors, compared with the reference, the risk of developing CKD was increased 9.83-fold, 50.67-fold, 28.96-fold, and 12.08-fold in the first, second, third, and fourth physical examinations, respectively, and differences among physical examinations were significant. After controlling for age, the results were consistent. 

## 4. Discussion

### 4.1. Epidemiology of CKD

#### 4.1.1. Prevalence of CKD in Our Air Force

According to a study conducted on the aircrew members in 2011, the prevalence of CKD was 10.4%, and the rates of stages 1–3 CKD were 6.45%, 3.72%, and 0.23%, respectively [3]. This study enrolled 212 aircrew members who could be tracked for four consecutive years, and the evaluation results by the aMDRD formula showed that the prevalence rate of CKD increased from 3.8% at the first physical examination to 9.4% at the fourth physical examination, and in each physical examination, only stages 1–3 CKD were found. The fourth physical examination showed that ten (4.70%) had stage 1 CKD, seven (3.29%) had stage 2 CKD, and three (1.41%) had stage 3 CKD. In this study, the prevalence of CKD is low, but the proportion of stage 3 CKD is higher than that in previous studies because the number of stage 3 CKD patients in this study is the same as that in the study by Zhang (both studies have only three stage 3 CKD patients), but the total sample size of this study is small, which resulted in a higher proportion. 

The grouping of the fourth physical examination results was performed according to age (the subjects in this physical examination were all older than 30 years); the prevalence of CKD in the 30–39-year age group was 12.0%, the prevalence of CKD in the 40–49-year age group was 9.8%, and no one in the over 50 age group had CKD. Since age is a known risk factor for CKD, previous studies have also shown that the overall prevalence of CKD increases with age [4], but the results of this study showed that the prevalence of CKD decreased with increasing age and that the incidence rate in the 30–39-year age group (12.0%) was the highest. A possible reason for this finding is that the subjects in this study are aircrew members, and thus, if they have a medical condition, they cannot continue to serve in the Air Force; in this study, early-stage CKD (stages 1 and 2) is dominant, and the GFR is still within the normal range; however, to diagnose early-stage CKD, renal parenchymal damage is required (at least one item of urine protein, urine RBC positivity, or urine pH > 8), thus reducing the explanatory power of age on the risk of early-stage CKD. 

Liu conducted a study on military personnel in the southern region in 2012 and reported that the overall prevalence of CKD was 14.6% and that the prevalence of CKD was 12.69%, 16.90%, and 14.29% for the 20–30-year age group, the 30–40-year age group, and the 40–60-year age group, respectively [5]. The large-scale study performed by Wen et al. showed that the overall prevalence of CKD was 11.93% and that the prevalence of CKD was 10.1%, 16.2%, and 23.6% for the 30–40-year age group, the 40–49-year age group, and the 50–60-year age group [6], which indicates that the prevalence of CKD in military personnel in the 30–40-year age group is significantly higher than that of ordinary individuals, and therefore, screening for early-stage CKD should be strengthened. 

#### 4.1.2. Incidence of CKD

Among the 212 valid samples in this study, 45 had CKD (minimum one time, maximum four times). With the first physical examination as the baseline and excluding 8 aircrew members who had CKD, 17 were found to have CKD in the second physical examination, with an incidence rate of 8.3%; 10 were found to have CKD in the third physical examination, with an incidence rate of 5.3%, and 10 were found to have CKD in the fourth physical examination, with an incidence rate of 5.6%. 

In summary, the incidence and prevalence of CKD in the second physical examination were higher than those in other physical examinations, and specifically, the differences in the results of the first physical examination (2007, 171 aircrew members) were the largest. An analysis revealed no changes in the aircraft type and unit, but the possibility of switching between aircrew and ground crew existed, the reasons for which are worthy of further research. The results of this study also showed that approximately 1/3 of the aircrew with CKD could continue to experience CKD, and thus, it is necessary to strengthen treatment and health education for this group of individuals and to strengthen early screening for CKD to avoid end-stage CKD, which could reduce the National Army’s military strength. 

#### 4.1.3. SCr Concentration

Patients with early-stage CKD may have either abnormal GFR or kidney pathological changes or both simultaneously [7], and SCr could be re-secreted by the renal tubules; therefore, the decline in GFR at the initial stage of CKD is not significant and only becomes severe when renal function deteriorates to stage 3 CKD [8,9]; that is, SCr within the normal range does not indicate normal renal function. In this study, SCr concentration was divided into quartiles (< 0.90 mg/dL, ≥ 0.90, and < 1.00 mg/dL, ≥ 1.00, < 1.06 mg/dL, and ≥ 1.06 mg/dL), and the results showed that FBG, urine BUN, and uric acid levels tended to increase with increasing SCr concentration (*p* < 0.05). Previous studies have shown that diabetes, cardiovascular disease, and high uric acid can lead to deterioration of renal function [10,11,12], but this study demonstrated that the prevalence rate of CKD was not linearly correlated with SCr concentration and was the highest in the second quartile. In patients with stage 1–2 CKD, the SCr concentration was normal, and the SCr concentration started to become abnormal only in stage 3 CKD, possibly because the GFR in the aircrew with stage 1–2 CKD was still within the normal range. According to the aMDRD formula, the SCr concentration is an important factor that affects GFR. The higher the SCr concentration, the higher the risk of developing CKD. Early-stage CKD can only be diagnosed by renal parenchymal damage (at least one item of urine protein, urine RBCs, or urine pH > 8), and thus, it can be inferred that for the diagnosis of early-stage CKD, aside from the GFR, the most important indication is renal parenchymal damage. Therefore, to effectively prevent and detect CKD at an earlier stage, regular routine urine and blood biochemical tests must be performed.

### 4.2. Discussion on Predictors of CKD

In this study, based on the consecutive four-year physical examination results of 212 valid aircrew members, the physical examination indicators that were significantly different between the CKD and non-CKD groups were used as variables in the GEE, and the first 10 variables with high QIC appropriateness were selected for analysis and comparison.

First, the correlation among the variables was discussed, and age, ketones in the urine, urine WBCs, FBG, BUN, SCr, and uric acid were negatively correlated with GFR, while urine occult blood and urine pH were positively correlated with GFR; these correlations were significant. To predict the risk of developing CKD, this study performed two analyses consisting of a univariate analysis and a year-by-year prediction analysis, which were conducted using multivariate logistic regression analysis. 

#### 4.2.1. Univariate Analysis

Based on the results of previous physical examinations, the comparison of the risk of CKD according to each variable showed that abnormal urobilinogen could increase the risk of developing CKD in all four physical examinations and that the OR in the first physical examination was the highest at 19.800 (95% CI: 3.819–102.659); urine occult blood was also found to increase the risk of developing CKD in three physical examinations, and the OR in the third physical examination was the highest at 10.175 (95% CI: 3.679–28.147); abnormal urine WBCs could increase the risk of developing CKD in three physical examinations, and the OR in the second physical examination was the highest at 14.973 (95% CI: 3.641–6.578). Among the risk factors for CKD, only abnormal urobilinogen and urine occult blood are reported in both this study and Zhang’s study on the same group in 2011, possibly because the number of early-stage CKD cases in this study is small and correlations among the variables are different. 

Previous studies have revealed many causes of CKD, the most common of which are hypertension and diabetes. Hypertension is associated with CKD, patient age, degree of renal function decline, proteinuria, and primary kidney disease [13] and is an important pathogenic factor for CKD. Hypertension not only affects the progression of renal disease in patients with CKD and accelerates the deterioration of renal function but also increases the risk of cardiovascular events [14,15]. According to statistics, approximately 40–50% of new cases of end-stage renal disease are caused by diabetes. In the past 10 years, the incidence and prevalence of kidney diseases caused by diabetes globally increased [16]. Dr. Fugate has also indicated that for diabetic patients, the risk of developing CKD could increase over time [17]. However, among the risk factors of CKD in the subjects in this study, hypertension and hyperglycaemia were not significantly correlated with CKD, possibly due to the low number of aircrew members with recorded BP data (only 55). Hyperglycaemia may be affected by the additional conditions of stage 1–2 CKD (meeting at least one item of urine protein, urine RBC positivity, or urine pH > 8) but was not significantly correlated with CKD. 

#### 4.2.2. Year-by-Year Prediction Analysis

To date, few studies have implemented long-term predictive analyses of CKD risk factors for special groups of individuals. One of the important aims of this study is to verify whether the risk of CKD in the next year, two years, and three years can be predicted by abnormal (CKD) indicators in one year’s physical examination so that CKD can be detected and treated at an early stage. The results showed that abnormal urobilinogen and urine occult blood in one year’s physical examination can predict the risk of CKD in the next year and that the maximum OR is 5.471 (95% CI: 1.255–23.845) and 8.860 (95% CI: 2.943–26.674), respectively. Abnormal urobilinogen, urine occult blood, abnormal urine WBCs, and abnormal BUN in one year’s physical examination can predict the risk of CKD in the next two years, while abnormal ketones in urine and urine occult blood in one year’s physical examination can predict the risk of CKD in the next three years, with maximum ORs of 5.004 (95% CI: 1.165–21.497) and 4.590 (95% CI: 1.441–14.614), respectively. CKD can also increase the risk of continuing to have CKD in the next year, with a maximum OR of 14.891 (95% CI: 5.022–44.152). 

The above analysis showed that urine occult blood can predict the risk of CKD in each model with statistical significance, with an R^2^ of 0.047–0.192. Urine containing RBCs is called urine occult blood and is often used clinically to indicate kidney diseases, such as renal trauma, kidney stones, renal tumours, and acute nephritis. Although urine occult blood does not directly indicate CKD, it has a high correlation with renal parenchymal damage [18]. Moreover, patients with CKD have a higher risk of continuing to have CKD. The other indicators also have predictive ability for CKD in different periods, but the R^2^ is quite different and is approximately 0.036–0.221. In this study, for the definition of early-stage CKD, in addition to considering the GFR, the detection of indicators of renal parenchymal damage, such as urine protein, urine RBC positivity, and urine pH > 8, is needed; Spearman’s correlation analysis showed that the above indicators were well correlated with some risk factors, such as urobilinogen, urine occult blood, urine WBCs, and ketones in the urine. Therefore, a long-term study on risk factors, such as urobilinogen, urine occult blood, urine WBCs, and ketones in the urine, can be conducted on the national population, to increase the accuracy and reliability of predicting the risk of early-stage CKD. 

### 4.3. Discussion of the Prediction Model for Long-Term Trends in CKD

Early detection and awareness of CKD can play an important role in future prognosis and in protecting kidney function from deterioration. Long-term trend research often requires multiple years. Yamagata et al. conducted a 10-year community-based study on 123,764 adults over the age of 40 and reported that men with hypertension and diabetes were twice as likely to have CKD than women, and men with proteinuria or hematuria had twice the risk of developing CKD ≥ stage 3 [19]. A follow-up study of 14,155 community residents found that age, sex, anaemia, hypertension, diabetes, heart failure, and cardiovascular disease were correlated with the development of kidney disease [20]. A large-scale cohort study of 462,293 individuals in Taiwan (with a mean follow-up time of 7.5 years) found that for Chinese medicine users, the risk of developing CKD was 1.2 times higher; if all stages of CKD were combined, the mortality rate of all causes was 1.83 higher, and the mortality due to cardiovascular disease was also twice as high [6].

Currently, few studies have been published on the establishment of CKD prediction models for special groups of people. Another important aim of this study was to use the GEE to establish a prediction model for the long-term trends in CKD. The results showed that the time (sequence) of physical examinations increased the risk of developing CKD and that the difference was significant. However, the length of service had a protective effect, which indicates that only healthy aircrew members can continue to serve in the Air Force. Urine occult blood, abnormal urobilinogen, abnormal urine WBCs, and abnormal ketones in the urine can all serve as independent predictors of CKD, and the difference is significant. Among the prediction model combinations, the QIC appropriateness of the prediction model using the combination of urine occult blood, urobilinogen, urine WBCs, ketones in the urine, and length of the service for the long-term trends in CKD is best, with a QIC of 395.418. The other combinations with good appropriateness are as follows: the prediction model combination of urine occult blood, urobilinogen, urine WBCs, and ketones in the urine, with a QIC of 397.441; the prediction model combination of urine occult blood, urobilinogen, and urine WBCs, with a QIC of 399.65; the prediction model combination of urine occult blood, urobilinogen, and ketones in the urine, with a QIC of 405.33; the prediction model combination of urine occult blood and urobilinogen, with a QIC of 408.94; and the prediction model combination of urine occult blood and urine WBCs, with a QIC of 418.18. These results are consistent with the results of the year-by-year prediction analysis, which further indicates the importance of each predictor for the risk of developing CKD. 

### 4.4. Discussion of CKD and the Number of Abnormal Predictors

The results of this study showed that the mean GFR decreased with age and was significantly different between physical examinations. For the aircrew with zero abnormal predictors (such as urine occult blood, urobilinogen, urine WBCs, and ketones in the urine), the negative predictive value for not having CKD was 96.65%; for the aircrew with one or two abnormal predictors, the positive predictive values of having CKD were 14.03% and 41.93%, respectively. According to the logistic regression analysis, the more abnormal the predictors, the higher the risk of developing CKD. With more than two abnormal predictors, the differences between physical examinations were significant. 

According to the definition of early-stage CKD, in addition to calculating the GFR, renal parenchymal damage is needed. In the study by Zhang, the mean change in the overall GFR was −3.79 mL/min/year [3]. In this study, for aircrew members over 50 years of age, the decrease in GFR between the first and second physical examinations was the greatest, and GFR was well maintained between the second and third physical examinations; for aircrew members in the 20–29-year age group, the decline in GFR became more severe; for all aircrew members, the decline in GFR between the third and fourth physical examinations was the more severe compared with declines in GFR in previous physical examinations. According to current policy, Air Force aircrews undergo one annual physical examination, the results of which are used as the basis for the continuation of aircrew service. According to the definition of CKD by the National Kidney Foundation and KDOQI clinical practice guidelines, the course of CKD is more than three months. As applicable, each guideline should be accompanied by background/rationale information, a detailed justification, close monitoring and evaluation guidance, implementation considerations, special discussions, and recommendations in evidence-based medicine [21,22,23]; therefore, the golden opportunity for treatment is often missed after diagnostic confirmation. According to the results of the prediction model in this study, concerning the predictors that have a high QIC appropriateness for CKD and that can be measured in a convenient, simple, and rapid manner (such as using test strips to detect urine occult blood, urobilinogen, ketones in the urine, urine WBCs, urine protein, and urine pH), if they can be monitored regularly, and if any abnormalities receive more attention, then the purpose of early detection and treatment of CKD can be realized, thus reducing damage to renal function.

### 4.5. Study Limitations

Since no data on serum cystatin C (Cys C) and eating (living) habits were collected, no analysis of those factors was performed.Due to the special operating environment and extremely strict screening of the Air Force aircrew, only healthy aircrew can continue to serve. Therefore, no stage 3B-5 CKD cases were included, and, therefore, it is necessary to be more cautious when inferring the research results.This study cannot confirm the renal parenchymal damage, such as urine protein, urine RBC positivity, and urine pH > 8, since subjects had early-stage CKD and cannot confirm whether the other chronic diseases such as chronic hepatitis B or C infection were associated with kidney diseases [24].Most of the subjects in this study are located in the north and south regions of Taiwan (accounting for 83%), which increases the difficulty of finding certain factors, and thus, the generalizability to other regions should be limited.In the Taiwan Air Force, only medical records with relatively healthy subjects could be involved. The airline service experience is defined as [age -26]. However, detailed total flight time, severity of gravity for bodies, and background diseases may have an impact on the prognosis for kidney diseases.

### 4.6. Research Suggestions and Future Prospects

The results of this study showed no obvious abnormality in SCr in Air Force aircrews with early-stage CKD; therefore, simply using SCr as a criterion for the diagnosis of CKD is likely to underestimate the number of CKD cases. For early-stage CKD, in addition to calculating the GFR, the determination of renal parenchymal damage is needed, and this study used urine protein, urine RBC positivity, and urine pH > 8 as the determination criteria. In addition, this study found that urine occult blood, urobilinogen, urine WBCs, and ketones in the urine can all predict the risk of developing CKD. On this basis, the feasibility of implementing simple and highly accurate preliminary tests (such as tests to determine urine occult blood, urobilinogen, ketones in the urine, urine WBCs, urine protein, and urine pH) in addition to the annual aircrew physical examination can be discussed, and even promotion at the nationwide level can also be discussed in the future to facilitate early detection and early treatment of CKD. 

Symptoms of CKD include kidney damage and decreased renal function, and thus, kidney damage must be considered. Many studies have found that Cys C is more sensitive to the decline in GFR than SCr and have suggested that Cys C can be used instead of SCr as an indicator of renal function, as the Cys C level can reflect both GFR and albuminuria in individual cases [8]. In addition, Shi also reported a high correlation between neck circumference and renal function indicators, such as GFR (r = 0.382), albuminuria (r = 0.304), and SCr (r = 0.181), and all correlations were significant [25]. Therefore, if Cys C and neck circumference can be added to annual aircrew physical examinations in the future, it might be helpful for the early detection of CKD and for the prevention and treatment of CKD. 

In the study on the same aircrew group, Zhang suggested that the physical examination report shows the GFR and the CKD stage for the self-management of health conditions by the aircrew [3]. Since the National Army has invested considerable resources in the training of aircrews, the health status of the aircrew is more related to the performance of the National Army’s military strength, and this relationship should not be underestimated. Therefore, this study recommended that for aircrews with abnormal CKD indicators, which can be revealed by a simple and accurate preliminary test using test strips or by the annual aircrew physical examination, the unit medical officer should regulate and assist the referral of the aircrew to a regional military hospital for further diagnosis and consultation with a specialist. This practice would help with compliance with the principle of early screening, real-time treatment, and continuous follow-up.

The CKD Epidemiology Collaboration (CKD-EPI) equation was developed in an effort to create a more precise formula for eGFR, especially if the actual GFR is more than 60 mL/min per 1.73 m^2^ [26,27]. Since the aMDRD formula for calculating GFR currently used in Taiwan is based on a Western population, this study suggests that in subsequent studies to validate the formula for assessing renal function, risk factors that are more suitable for the Taiwanese population should be added to establish a local formula that is more compatible with this population. In addition, previous study results demonstrated that combined albuminuria and neck circumference could provide accurate eGFR predictions in patients with cardiovascular disease [28]. Neck circumference and associated physical status may be investigated potentially in further studies [29,30,31].

## 5. Conclusions

Worldwide, the incidence and prevalence of CKD are rapidly increasing and have attracted much attention from public health and medical circles. The incidence and prevalence of end-stage renal disease in Taiwan have ranked first in the world in multiple years. Since the treatment of end-stage renal disease consumes many healthcare resources, and the early symptoms of CKD are not obvious, most people are not aware of CKD, and, therefore, understanding the factors associated with early-stage CKD and taking the necessary preventive measures and health management at an early time can reduce the risk of developing CKD in military personnel. 

In this study, logistic regression was used to analyse the predictors of CKD. After controlling for age, abnormal urobilinogen, abnormal ketones in the urine, abnormal urine WBCs, and urine occult blood were found to increase the risk of developing CKD, and this risk increases with the increase in the number of abnormal predictors. In terms of long-term predictive ability, urine occult blood can predict the risk of CKD in the next year, two years, and three years with statistical significance, with R^2^ values of 0.128, 0.186, and 0.063, respectively. In terms of the prediction model for the long-term trends in CKD, the GEE was used to examine these trends under repeated measurements over time, and the results showed that the time (sequence) of physical examinations can increase the risk of developing CKD, whereas the length of service had a protective effect, and the differences were all significant. Urine occult blood, abnormal urobilinogen, abnormal urine WBCs, and abnormal ketones in the urine can all serve as independent predictors of CKD, and the differences were all significant.

## Figures and Tables

**Figure 1 bioengineering-11-00231-f001:**
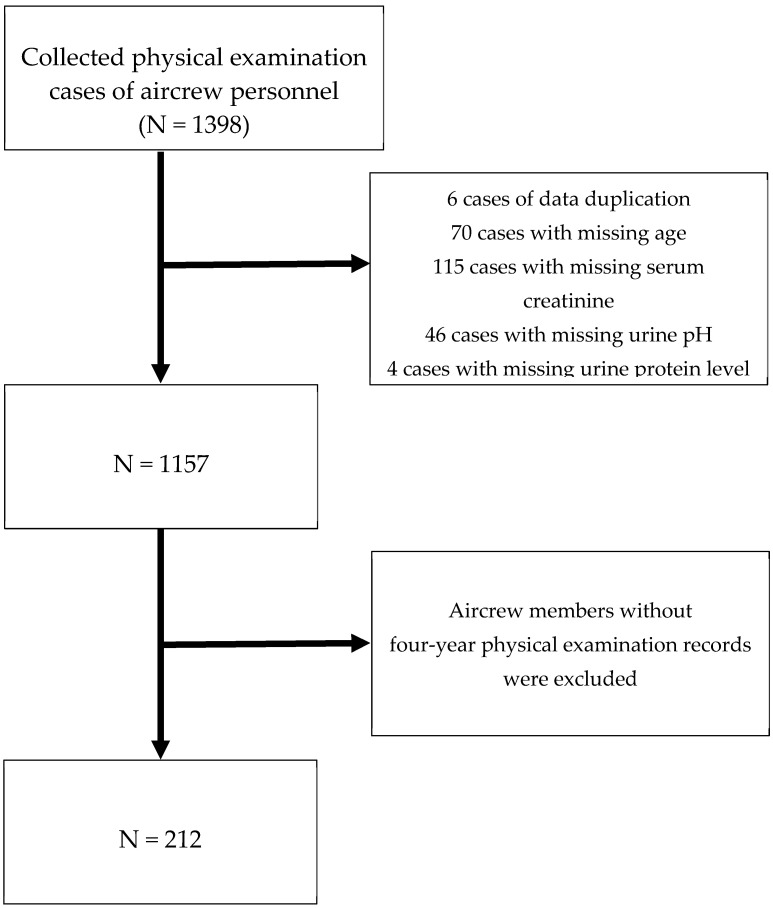
Study flow chart.

**Table 1 bioengineering-11-00231-t001:** Comparison of CKD correlation in previous physical examinations (N = 212).

Item	Group	First Physical Examination		Second Physical Examination		Third Physical Examination		Fourth Physical Examination		*p* Value	*p* for Trend
		Number of People ^a^ (%)	Mean ± SD	Number of People ^a^ (%)	Mean ± SD	Number of People ^a^ (%)	Mean ± SD	Number of People ^a^ (%)	Mean ± SD		
eGFR			107.29 ± 19.34		103.58 ± 17.34		102.37 ± 16.84		93.14 ± 16.01	<0.001	<0.001
(mL/min/1.73 m^2^)											
	≥90	171 (80.7)		164 (77.4)		161 (75.9)		119 (56.1)		<0.001	
	60–89	41 (19.3)		48 (22.6)		50 (23.6)		90 (42.5)			
	45–59	0 (0)		0 (0)		1 (0.5)		3 (1.4)			
CKD											
	non-CKD	204 (96.2)		192 (90.6)		193 (91.0)		192 (90.6)		0.084	
	CKD	8 (3.8)		20 (9.4)		19 (9.0)		20 (9.4)			
CKD stage											
	Stage 1	8 (3.8)		14 (6.58)		11 (5.21)		10 (4.7)		0.136	
	Stage 2	0 (0)		6 (2.82)		7 (3.32)		7 (3.29)			
	Stage 3	0 (0)		0 (0)		1 (0.47)		3 (1.41)			

^a^ If the number of people for one item is not equal to the total number, people are missing for this item, and thus, the effective percentage is presented. Continuous variable: the *p* value is obtained by analysis of variance (ANOVA) and linear regression; categorical variable: the *p* value is obtained by the two-tailed chi-square test.

**Table 2 bioengineering-11-00231-t002:** Comparison of previous physical examination indicators and CKD risk in the Air Force (N = 212).

Item	First Physical Examination		Second Physical Examination		Third Physical Examination		Fourth Physical Examination	
	Odds Ratio (OR) (95% CI)	Adjusted OR (95% CI) ^a^	OR (95% CI)	Adjusted OR (95% CI) ^a^	OR (95% CI)	Adjusted OR (95% CI) ^a^	OR (95% CI)	Adjusted OR (95% CI) ^a^
Age	0.910 (0.804–1.030)		0.940 (0.874–1.011)		0.954 (0.888–1.025)		0.963 (0.899–1.032)	
Abnormal urobilinogen (vs. normal)	19.800 (3.819–102.659) **	16.969 (3.158–91.189) *	6.067 (1.835–20.055) *	4.820 (1.362–17.053) *	8.259 (2.384–28.609) *	7.414 (2.091–26.290) *	4.590 (1.441–14.614) *	4.359 (1.358–13.986) *
Abnormal ketones in urine (vs. normal)	3.095 (0.343–27.914)	3.762 (0.395–35.798)	2.950 (0.875–9.951)	3.806 (1.065–13.601) *	4.880 (1.366–17.436) *	5.301 (1.447–19.416) *	1.222 (0.260–5.747)	1.352 (0.283–6.457)
Urine occult blood detected (vs. normal)	1.571 (0.182–13.535)	1.331 (0.153–11.583)	10.000 (3.435–29.111) **	8.712 (2.914–26.043) **	10.175 (3.679–28.147) **	9.539 (3.383–26.900) **	6.444 (2.329–17.831) **	6.113 (2.187–17.085) *
Abnormal urine WBCs (vs. normal)	-	-	12.467 (3.243–47.924) **	14.973 (3.641–61.578) **	2.400 (0.796–7.240)	3.387 (1.025–11.186) *	4.059 (0.984–16.739)	4.324 (1.033–18.094) *
High FBG (vs. normal)	1.933 (0.373–10.032)	3.139 (0.547–18.011)	1.041 (0.381–2.845)	1.330 (0.468–3.783)	0.563 (0.180–1.768)	0.659 (0.204–2.132)	0.301 (0.085–1.062)	0.319 (0.090–1.135)
Abnormal BUN (vs. normal)	-	-	3.316 (0.329–33.465)	3.609 (0.350–37.207)	-	-	3.316 (0.329–33.465)	2.848 (0.273–29.707)
Abnormal SCr (vs. normal)	-	-	-	-	1.731 (0.197–15.188)	1.863 (0.210–16.526)	1.706 (0.456–6.385)	1.842 (0.486–6.986)
Abnormal uric acid (vs. normal)	2.091 (0.057–8.621)	2.042 (0.491–8.498)	0.973 (0.336–2.816)	0.863 (0.295–2.526)	0.406 (0.114–1.445)	0.391 (0.109–1.399)	1.351 (0.462–3.948)	1.278 (0.434–3.760)
Abnormal LDLC (vs. normal)	5.500 (0.629–48.125)	7.885 (0.843–73.774)	2.286 (0.758–6.890)	2.126 (0.694–6.514)	1.005 (0.390–2.588)	1.128 (0.429–2.965)	0.420 (0.147–1.201)	0.441 (0.153–1.271)

^a^ Multivariate analysis after controlling for age. * *p* < 0.05, ** *p* < 0.001. If no OR is presented, the variable is a constant since no statistics can be calculated. R^2:^ Only significant values are listed. (1) First physical examination: urobilinogen 0.164, 0.190. (2) Second physical examination: urobilinogen 0.073, 0.083; ketones in urine 0.067; urine occult blood 0.154, 0.163; urine WBCs 0.116, 0.154. (3) Third physical examination: urobilinogen 0.094, 0.101; ketones in urine 0.051, 0.072; urine occult blood 0.187, 0.190; urine WBCs 0.055. (4) Fourth physical examination: urobilinogen 0.056, 0.064; urine occult blood 0.113, 0.117; urine WBCs 0.045.

**Table 3 bioengineering-11-00231-t003:** Air Force aircrew physical examination indicators and long-term prediction models (1–3) (**a**) and (4–6) (**b**) of CKD risks (N = 212).

(**a**)						
**Item**	**Model 1**		**Model 2**		**Model 3**	
	**OR (95% CI)**	**Adjusted OR (95% CI) ^a^**	**OR (95% CI)**	**Adjusted OR (95% CI) ^a^**	**OR (95% CI)**	**Adjusted OR (95% CI) ^a^**
Abnormal urobilinogen (vs. normal)	5.471 (1.255–23.845) *	4.771 (1.068–21.317) *	2.828 (0.723–11.069)	2.265 (0.544–9.441)	5.083 (1.408–18.353) *	4.618 (1.250–17.060) *
Abnormal ketones in urine (vs. normal)	1.070 (0.129–8.910)	1.176 (0.139–9.950)	1.218 (0.259–5.724)	1.411 (0.293–6.794)	1.667 (0.346–8.038)	1.742 (0.358–8.477)
Urine occult blood detected (vs. normal)	8.860 (2.943–26.674) **	7.942 (2.613–24.139) **	8.077 (2.719–23.997) **	7.366 (2.398–22.625) **	4.161 (1.500–11.545) *	3.875 (1.372–10.941) *
Abnormal urine WBCs (vs. normal)	1.632 (0.186–14.275)	1.784 (0.199–16.007)	2.721 (0.535–13.846)	2.890 (0.559–14.946)	1.078 (0.296–3.930)	1.313 (0.343–5.030)
High FBG (vs. normal)	1.464 (0.456–4.702)	2.034 (0.594–6.967)	0.852 (0.293–2.477)	1.015 (0.338–3.055)	0.222 (0.050–0.987) *	0.241 (0.053–1.096)
Abnormal BUN (vs. normal)	0.866 (0.106–7.09)	0.919 (0.111–7.628)	3.519 (0.348–35.597)	3.711 (0.362–38.021)	1.632 (0.186–14.275)	1.665 (0.189–14.712)
Abnormal SCr (vs. normal)	-	-	-	-	1.632 (0.186–14.275)	1.721 (0.195–15.176)
Abnormal uric acid (vs. normal)	0.857 (0.315–2.335)	0.830 (0.302–2.276)	0.147 (0.019–1.127)	0.130 (0.017–1.007)	0.990 (0.363–2.704)	0.966 (0.353–2.647)
Abnormal LDLC (vs. normal)	1.867 (0.617–5.648)	1.523 (0.471–4.927)	1.089 (0.371–3.195)	1.044 (0.349–3.125)	0.482 (0.176–1.324)	0.513 (0.909–1.048)
CKD (vs. non-CKD)	6.600 (1.451–30.027) *	5.565 (1.194–25.950) *	14.891 (5.022–44.152) **	13.958 (4.659–41.816) **	8.077 (2.719–23.997) **	7.695 (2.576–22.982) **
(**b**)						
**Item**	**Model 4**		**Model 5**		**Model 6**	
	**OR (95% CI)**	**Adjusted OR (95% CI) ^a^**	**OR (95% CI)**	**Adjusted OR (95% CI) ^a^**	**OR (95% CI)**	**Adjusted OR (95% CI) ^a^**
Abnormal urobilinogen (vs. normal)	3.126 (0.602–16.246)	2.772 (0.523–14.684)	1.211 (0.144–10.205)	1.081 (0.126–9.250)	6.067 (1.835–20.055) *	5.613 (1.568–20.090) *
Abnormal ketones in urine (vs. normal)	2.271 (0.535–13.846)	2.946 (0.569–15.250)	4.664 (1.104–19.701) *	5.004 (1.165–21.497) *	2.950 (0.875–9.951)	3.443 (0.985–12.036)
Urine occult blood detected (vs. normal)	13.309 (4.381–40.429) **	12.347 (4.018–37.943) **	4.590 (1.441–14.614) *	4.252 (1.320–13.696) *	3.933 (1.257–12.313) *	3.568 (1.107–11.496) *
Abnormal urine WBCs (vs. normal)	-	-	1.632 (0.186–14.275)	1.723 (0.195–15.258)	4.664 (1.104–19.701) *	4.929 (1.149–21.141) *
High FBG (vs. normal)	0.639 (0.140–2.911)	0.777 (0.165–3.668)	0.600 (0.132–2.721)	0.698 (0.149–3.268)	0.578 (0.185–1.803)	0.650 (0.202–2.089)
Abnormal BUN (vs. normal)	0.919 (0.112–7.533)	0.961 (0.116–7.951)	3.588 (0.887–14.521)	3.777 (0.921–15.494)	10.556 (1.402–79.467) *	11.071 (1.453–84.366) *
Abnormal SCr (vs. normal)	-	-	2.259 (0.453–11.267)	2.412 (0.478–12.174)	5.000 (0.433–57.730)	5.335 (0.451–63.097)
Abnormal uric acid (vs. normal)	0.353 (0.099–1.253)	0.341 (0.096–1.218)	0.857 (0.315–2.335)	0.839 (0.307–2.292)	1.028 (0.355–2.976)	0.898 (0.307–2.625)
Abnormal LDLC (vs. normal)	1.205 (0.404–3.597)	1.050 (0.328–3.365)	1.458 (0.470–4.524)	1.124 (0.334–3.789)	2.503 (0.778–8.057)	2.310 (0.706–7.559)
CKD (vs. non-CKD)	1.476 (0.172–12.678)	1.250 (0.143–10.945)	3.444 (0.647–18.331)	3.059 (0.564–16.597)	13.463 (4.614–39.279) **	12.860 (4.352–37.997) **

Model 1: Prediction of CKD risk in the second physical examination using indicators from the first physical examination. Model 2: Prediction of CKD risk in the third physical examination using indicators from the second physical examination. Model 3: Prediction of CKD risk in the fourth physical examination using indicators from the third physical examination. Model 4: Prediction of CKD risk in the third physical examination using indicators from the first physical examination. Model 5: Prediction of CKD risk in the fourth physical examination using indicators from the first physical examination. Model 6: Prediction of CKD risk in the fourth physical examination using indicators from the second physical examination. ^a^ Multivariate analysis after controlling for age. * *p* < 0.05, ** *p* < 0.001. If no OR is presented, the variable is a constant since no statistics can be calculated. R^2^: Only significant values are listed. (1) Model 1: urobilinogen 0.042, 0.066; urine occult blood 0.128, 0.146; CKD 0.049, 0.072. (2) Model 2: urine occult blood 0.124, 0.127; CKD 0.216, 0.221. (3) Model 3: urobilinogen 0.051, 0.059; urine occult blood 0.066, 0.071; FBG 0.055; CKD 0.121, 0.127. (4) Model 4: urine occult blood 0.186, 0.192. (5) Model 5: ketones in urine 0.036, 0.051; urine occult blood 0.056, 0.063. (6) Model 6: urobilinogen 0.073, 0.074; urine occult blood 0.047, 0.052; urine WBCs 0.036, 0.050; BUN 0.045, 0.058; CKD 0.200, 0.202.

**Table 4 bioengineering-11-00231-t004:** Comparison of long-term trends in CKD and prediction models (1–5) (**a**) and (6–9) (**b**) among the Air Force aircrew (N = 212).

(**a**)											
**Variable**		**Model 1**		**Model 2**		**Model 3**		**Model 4**		**Model 5**	
		**Coefficient**	**(CF)OR (95% CI)**	**CF**	**OR (95% CI)**	**CF**	**OR (95% CI)**	**CF**	**OR (95% CI)**	**CF**	**OR (95% CI)**
Physical examination time (sequence)											
	First physical examination (reference)										
	Second physical examination	0.977	2.656 (1.234–5.716) *	0.998	2.712 (1.172–6.272) *	0.933	2.543 (1.166–5.548) *	0.919	2.507 (1.162–5.412) *	0.921	2.513 (1.167–5.409) *
	Third physical examination	0.920	2.510 (1.084–5.812) *	0.828	2.289 (0.942–5.558)	0.862	2.367 (1.022–5.484) *	0.708	2.030 (0.853–4.834)	0.892	2.439 (1.051–5.664) *
	Fourth physical examination	0.977	2.656 (1.192–5.921) *	0.955	2.599 (1.113–6.071) *	0.891	2.438 (1.098–5.418) *	0.915	2.497 (1.116–5.588) *	0.940	2.560 (1.151–5.692) *
Predictor											
	Urine occult blood (vs. normal)			1.909	6.746 (3.383–13.456) **						
	Abnormal urobilinogen (vs. normal)					1.725	5.614 (3.021–10.433) **				
	Abnormal urine WBCs (vs. normal)							1.336	3.805 (1.996–7.253) **		
	Abnormal ketones in urine (vs. normal)									1.012	2.571 (1.498–5.054) *
(**b**)											
**Variable**		**Model 6**				**Model 7**		**Model 8**		**Model 9**	
		**CF**		**OR (95% CI)**		**CF**	**OR (95% CI)**	**CF**	**OR (95% CI)**	**CF**	**OR (95% CI)**
Physical examination time (sequence)											
	First physical examination (reference)										
	Second physical examination	0.942		2.565 (0.918–7.165)		1.019	2.770 (1.264–6.070) *	0.982	2.669 (1.230–5.793) *	0.980	2.665 (1.251–5.674) *
	Third physical examination	0.227		1.255 (0.428–3.674)		0.967	2.630 (1.121–6.172) *	0.926	2.523 (1.085–5.866) *	0.921	2.512 (1.090–5.791) *
	Fourth physical examination	0.230		1.258 (0.460–3.439)		1.040	2.829 (1.239–6.458) *	0.984	2.674 (1.186–6.029) *	0.981	2.668 (1.207–5.897) *
Predictor											
	Abnormal LDLC (vs. normal)	0.088		1.092 (0.647–1.843)							
	High FBG (vs. normal)					−0.341	0.711 (0.363–1.392)				
	Abnormal BUN (vs. normal)							0.178	1.195 (0.500–2.854)		
	Abnormal uric acid (vs. normal)									0.040	1.041 (0.514–2.107)

Dependent variable: with or without CKD. * *p* < 0.05, ** *p* < 0.001. Model 1: time and long-term changes in CKD (QIC: 469.00). Model 2: time, urine occult blood, and long-term changes in CKD (QIC: 428.43). Model 3: time, urobilinogen, and long-term changes in CKD (QIC: 441.68). Model 4: time, urine WBCs, and long-term changes in CKD (QIC: 458.98). Model 5: time, ketones in urine, and long-term changes in CKD (QIC: 464.62). Model 6: time, LDLC, and long-term changes in CKD (QIC: 400.82). Model 7: time, FBG, and long-term changes in CKD (QIC: 469.03). Model 8: time, BUN, and long-term changes in CKD (QIC: 471.00). Model 9: time, uric acid, and long-term changes in CKD (QIC: 471.07).

**Table 5 bioengineering-11-00231-t005:** (**a**) Comparison of CKD (**a**)/CKD Risk (**b**) and the number of abnormal predictors in the Air Force aircrew (N =212).

(**a**)																
**Item**	**First Physical Examination**				**Second Physical Examination**				**Third Physical Examination**				**Fourth Physical Examination**			
	**Non-CKD (n = 204)**	**CKD (n = 8)**		***p* Value**	**Non-CKD (n = 192)**	**CKD (n = 20)**		***p* Value**	**Non-CKD (n = 193)**	**CKD (n = 19)**		***p* Value**	**Non-CKD (n = 192)**	**CKD (n = 20)**		***p* Value**
	**Number of People (%)**	**Number of People (%)**			**Number of People (%)**	**Number of People (%)**			**Number of People (%)**	**Number of People (%)**			**Number of People (%)**	**Number of People (%)**		
Number of abnormal predictors																
0	172 (97.2)	5 (2.8)		0.012	152 (96.2)	6 (3.8)		<0.001	139 (97.2)	4 (2.8)		<0.001	145 (96.0)	6 (4.0)		<0.001
1	25 (96.2)	1 (3.8)			38 (82.6)	8 (17.4)			47 (85.5)	8 (14.5)			39 (79.6)	10 (20.4)		
2	7 (77.8)	2 (22.2)			2 (33.3)	4 (66.7)			6 (54.5)	5 (45.5)			8 (66.7)	4 (33.3)		
3	-	-			0 (0)	2 (100.0)			1 (33.3)	2 (66.7)			-	-		
(**b**)																
**Item**	**First Physical Examination**				**Second Physical Examination**				**Third Physical Examination**				**Fourth Physical Examination**			
	**OR (95% CI)**		**Adjusted OR (95% CI) ^a^**		**OR (95% CI)**		**Adjusted OR (95% CI) ^a^**		**OR (95% CI)**		**Adjusted OR (95% CI) ^a^**		**OR (95% CI)**		**Adjusted OR (95% CI) ^a^**	
Number of abnormal predictors																
0	1		1		1		1		1		1		1		1	
1	1.38 (0.154–12.266)		1.21 (0.134–10.939)		5.33 (1.746–16.290) *		5.18 (1.667–15.707) *		5.91 (1.703–20.541) *		5.97 (1.706–20.924) *		6.197 (2.121–18.103) *		5.831 (1.982–17.154) *	
2	9.83 (1.616–59.792) *		10.00 (1.575–63.544) *		50.67 (7.706–333.123) **		43.60 (6.491–292.878) **		28.96 (6.158–136.174) **		31.64 (6.446–155.293) **		12.08 (2.830–51.598) *		13.00 (2.985–56.669) *	
3	-		-		-		-		69.50 (5.172–933.965) *		105.84 (6.480–1728.642) *		-		-	

If no number is presented, the variable is a constant since no statistics can be calculated. Categorical variable: the *p* value is obtained by a two-tailed chi-square test. ^a^ Multivariate analysis after controlling for age. * *p* < 0.05, ** *p* < 0.001. If no OR is presented, the variable is a constant since no statistics can be calculated.

## Data Availability

The data underlying this article will be shared upon reasonable request to the corresponding author.

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
