# Peer review of "Biomedical Evaluation of Early Chronic Kidney Disease in the Air Force: Building a Predictive Model from the Taiwan Military Health Service"

_bioengineering, 2024, doi:10.3390/bioengineering11030231_

Round 1

Reviewer 1 Report

Comments and Suggestions for Authors

In the present article, authors presented demographic data to discuss progression of CKD in Twainian air force-crews. Although analysis did not give new aspects as general CKD patients, only advantage might be that influences of frequent changes of atmospheric pressueure and of severe gravity effects in body. However, their statistical analysis might not be satisfied for readers. Some data should be adjusted before evaluation. How does total flight tine, severity of gravity for bodies, and back ground diseases influenced for prognosis for renal survival? Text might be too long and messy. I recommend to reorganize the whole text body. In addition, there are also several comments which are listed as below.

1.        Most of tables may be demographic data to occupy the text body. Therefore, please compress the data.

2.        In Table 5, creatinine is essential to be adjusted with age, sex, BMI, and so on. Or, eGFR would be better to use instead of creatinine.

3.        It would better to use baseline data to compare the second and the later.

Comments on the Quality of English Language

NA

Author Response

Response to Reviewer 1

[General Comment]

In the present article, authors presented demographic data to discuss progression of CKD in Twainian air force-crews. Although analysis did not give new aspects as general CKD patients, only advantage might be that influences of frequent changes of atmospheric pressure and of severe gravity effects in body. However, their statistical analysis might not be satisfied for readers. Some data should be adjusted before evaluation. How does total flight time, severity of gravity for bodies, and back ground diseases influenced for prognosis for renal survival? Text might be too long and messy. I recommend to reorganize the whole text body. In addition, there are also several comments which are listed as below.

Author Reply: We sincerely appreciate your time and effort spent reviewing this manuscript. We have revised the manuscript thoroughly according to your suggestions. We reorganized the whole text body (from total pages 48 to 32). In Taiwan Air Force, only medical record with relative healthy subjects could be involved. The airline service experience is defined as (age-26). However, total flight time, severity of gravity for bodies, and back ground diseases may have the impact on the prognosis for kidney diseases. We added this point in the section of study limitations. The responses to your comments are found below. Please see page 29, lines 925-928.

4.5. Study Limitations

  1. This study used the data collected by the Retrospective Generational Tracking Research Plan of the Physical Examination Indicators of Taiwanese Military Aircrew. Since no data on serum cystatin C (Cys C), and eating (living) habits were collected, no analysis of those factors was performed.
  2. Due to the special operating environment and extremely strict screening of Air Force aircrew, only healthy aircrew can continue to serve. Therefore, no stage 3B-5 CKD cases were included, and therefore, it is necessary to be more cautious when inferring the research results.
  3. This study cannot confirm the renal parenchymal damage, such as urine protein, urine RBC positivity, and urine pH >8, since subjects had early-stage CKD and cannot confirm whether the other chronic diseases such as chronic hepatitis B or C infection associated kidney diseases [25].
  4. Most of the subjects in this study are located in the north and south regions of Taiwan (accounting for 83%), which increases the difficulty of finding certain factors, and thus, the generalizability to other regions should be limited.
  5. In Taiwan Air Force, only medical record with relative healthy subjects could be involved. The airline service experience is defined as (age-26). But detailed total flight time, severity of gravity for bodies, and back ground diseases may have the impact on the prognosis for kidney diseases.

  1. Most of tables may be demographic data to occupy the text body. Therefore, please compress the data.

Author Reply: Thank you for your valuable comments. We made this correction immediately. We shortened and reorganized the whole text body (from total pages 48 to 32). In addition, the demographic data including Table 1 to Table 7 were categorized into supplementary Tables 1-7.

  1. In Table 5, creatinine is essential to be adjusted with age, sex, BMI, and so on. Or, eGFR would be better to use instead of creatinine

Author Reply: Thank you for your valuable comments. We made this correction. In our study, CKD is defined as the presence of kidney damage based on the estimated glomerular filtration rate (eGFR) less than 60 ml/min/1.73m2, or evident proteinuria/haematuria, persisting for more than 3 months. Therefore, we used creatinine to be adjusted with age, sex, BMI, and so on. The results of the basic examination, routine urine test, and blood biochemical test were analysed according to the SCr quartiles (the first quartile is <0.90 mg/dl, the second quartile is ≥0.90 and <1.00 mg/dl, the third quartile is ≥1.00 and <1.06 mg/dl, and the fourth quartile is ≥1.06 mg/dl). Please see page 3, lines 111-119; page 6, 269-272, and supplementary Table 5.

  1. It would better to use baseline data to compare the second and the later.

Author Reply: Thank you for your valuable comments. In this study, we included all subjects at a military hospital from baseline (first year) and who could be tracked for four years. Comparison of long-term trends in CKD and prediction models among Air Force aircrew were performed. Please see Figure 1 and Tables 1-5.

Last, we are deeply honored by the time and effort you spent reviewing this manuscript. In reviewing and revising our manuscript, we are motivated to read more and thus learn more from your criticisms.

Reviewer 2 Report

Comments and Suggestions for Authors

Major revisions to be made to work with this manuscript:

1.      Study design should be described as cross-sectional or period prevalence.

2.      Were the subjects consented for the study?

3.      Do you have the recruitment flow chart for the clinic?

4.      How did the researchers define Chronic Kidney disease?

5.      Why was MDRD equation applied and not CKD epi (which is more appropriate for the study design)?

6.      The physical examination has different parts based on diagnoses to be considered as a predictor for CKD diagnosis.  Please break down the different physical examination signs into categories of diagnoses and then apply to the equation. 

7.      Similarly for urinalysis, certain cut offs of hematuria and pyuria shows clinical significance, please apply those definitions for prediction of CKD

8.      The correlation matrix supports the arguments above that a general statement of abnormal test is not specific enough for a predictor of an outcome, ie. CKD.

9.      The knowledge of certain signs and symptoms that predict CKD diagnosis are not described as separate entities for prediction.  Why was medical history information not included?

10.  The prevalence in a healthy general population for CKD should be very low.   Why is the number still detectable?  How long were these individuals in the airforce and what is the change in the study variables from baseline physical examination?

Comments on the Quality of English Language

A english writer should review the paper

Author Response

Response to Reviewer 2

[General Comment]

Major revisions to be made to work with this manuscript:

Author Reply: We sincerely appreciate your time and effort spent reviewing this manuscript. We have revised the manuscript thoroughly according to your suggestions. The responses to your comments are found below.

  1. Study design should be described as cross-sectional or period prevalence.

Author Reply: Thank you for your valuable comments. This study is a period prevalence study. We made this correction immediately. Please see page 2, lines 86-89.

This study is a period prevalence study. First, the research subject is defined, after which the research structure, research hypothesis, and operational definition of the research variables are described according to the literature and data; finally, data collection, data processing, and data analysis are described. The contents are as follows:…………

  1.  Were the subjects consented for the study?

Author Reply: Thank you for your valuable comments. Informed consent was obtained from all participants involved in this study. Please see page 3, lines 107-108.

  1. Do you have the recruitment flow chart for the clinic?

Author Reply: Thank you for your valuable comments. We added Figure 1 (study flow chart). Please see page 4, lines 151-193.

  1. How did the researchers define Chronic Kidney disease?

Author Reply:

Thank you for your valuable comments. To prevent overdiagnosis, CKD is defined as the presence of kidney damage based on the estimated glomerular filtration rate (eGFR) less than 60 ml/min/1.73m2, or evident proteinuria/haematuria, persisting for more than 3 months. In this study, the eGFR was according to Abbreviated Modification of Diet in Renal Disease Formula (aMDRD):  eGFR = 186 × Creatinine−1.154 × Age−0.203. We made this correction and added the reference 2 in the Method section. Please see page 3, lines 111-119.

Reference

  • Levey AS, Coresh J, Balk E, et al. National Kidney Foundation practice guidelines for chronic kidney disease: evaluation, classification, and stratification. Ann Intern Med 2003;139:137–47.
  1. Why was MDRD equation applied and not CKD epi (which is more appropriate for the study design)?

Author Reply:

Thank you for your valuable comments. The CKD Epidemiology Collaboration (CKD EPI) equation was developed in an effort to create a more precise formula to eGFR, especially actual GFR is more than 60mL/min per 1.73m2. Since the aMDRD formula for calculating GFR currently used in Taiwan is based on a western population, this study suggests that in subsequent studies to validate the formula for assessing renal function, risk factors that are more suitable for the Taiwanese population should be added to establish a local formula that is more compatible with this population. We made this correction and added the references (26,27) in the section of Future Prospects. Please see page 30, lines 963-969.

Reference

  1. Delanaye P, Cavalier E, Pottel H, Stehlé T. New and old GFR equations: a European perspective. Clin Kidney J. 2023 Mar 15;16(9):1375-1383. doi: 10.1093/ckj/sfad039.
  2. Miller WG, Kaufman HW, Levey AS, Straseski JA, Wilhelms KW, Yu HE, Klutts JS, Hilborne LH, Horowitz GL, Lieske J, Ennis JL, Bowling JL, Lewis MJ, Montgomery E, Vassalotti JA, Inker LA. National Kidney Foundation Laboratory Engagement Working Group Recommendations for Implementing the CKD-EPI 2021 Race-Free Equations for Estimated Glomerular Filtration Rate: Practical Guidance for Clinical Laboratories. Clin Chem. 2022 Mar 31;68(4):511-520. doi: 10.1093/clinchem/hvab278.

  1. The physical examination has different parts based on diagnoses to be considered as a predictor for CKD diagnosis.  Please break down the different physical examination signs into categories of diagnoses and then apply to the equation.

Author Reply: Thank you for your valuable comments. The study subjects’ data included the following: basic personal information (including date of birth, sex, unit location, and annual physical examination date; the name and identity card number were replaced by serial numbers) and annual aircrew physical examination indicators (including height, weight, blood pressure (BP), physical examination, and laboratory tests) to facilitate the confirmation of CKD. Supplementary Table 5-1 demonstrated the comparison of different serum creatinine (SCr) concentrations and basic physical examinations. In our previous study, we found neck circumference could be investigated potentially in kidney function. We made the corrections and added the references (28-31) in the section of our Future Prospects. Please see page 3, lines 111-119 and page 30, lines 970-972; page 6, lines 268-303.

Reference 

  1. Hsiao PJ, Lin HC, Chang ST, Hsu JT, Lin WS, Chung CM, Chang JJ, Hung KC, Shih YW, Chen FC, Hu FK, Wu YS, Chang CW, Su SL, Chu CM. Albuminuria and neck circumference are determinate factors of successful accurate estimation of glomerular filtration rate in high cardiovascular risk patients. PLoS One. 2018 Feb 2;13(2):e0185693. doi: 10.1371/journal.pone.0185693.
  2. Yoon Y, Kim YM, Lee S, Shin BC, Kim HL, Chung JH, Son M. Association between Neck Circumference and Chronic Kidney Disease in Korean Adults in the 2019-2021 Korea National Health and Nutrition Examination Survey. Nutrients. 2023 Dec 8;15(24):5039. doi: 10.3390/nu15245039.
  1. Huang HY, Lin TW, Hong ZX, Lim LM. Vitamin D and Diabetic Kidney Disease. Int J Mol Sci. 2023 Feb 13;24(4):3751. doi: 10.3390/ijms24043751.
  1. Hwang JS, Hwang IC, Ahn HY. The snoring, tiredness, observed apnea, high BP, BMI, age, neck circumference, and male gender (STOP-BANG) score and kidney function in a general population. J Nephrol. 2024 Jan 11. doi: 10.1007/s40620-023-01854-y. 

  1.  Similarly for urinalysis, certain cut offs of hematuria and pyuria shows clinical significance, please apply those definitions for prediction of CKD.

Author Reply: Thank you for your valuable comments. The study subjects’ data included the following: basic personal information (including date of birth, sex, unit location, and annual physical examination date; the name and identity card number were replaced by serial numbers) and annual aircrew physical examination indicators (including height, weight, blood pressure (BP), physical examination, and laboratory tests) to facilitate the confirmation of CKD. The normal ranges of urine RBC:0-2/HPF and urine WBC: 0-5/HPF were defined in this study. We made the different prediction models of CKD. Please see Table 3 and Table 4. Please see pages 16-20.

  1. The correlation matrix supports the arguments above that a general statement of abnormal test is not specific enough for a predictor of an outcome, ie. CKD.

Author Reply: Thank you for your valuable comments. The correlation matrix supports the arguments above that a general statement of abnormal test is indeed not specific enough for a predictor of an outcome of CKD. We removed this matrix according to your suggestions.

  1.  The knowledge of certain signs and symptoms that predict CKD diagnosis are not described as separate entities for prediction.  Why was medical history information not included?

Author Reply: Thank you for your valuable comments. CKD is defined as the presence of kidney damage based on the estimated glomerular filtration rate (eGFR) less than 60 ml/min/1.73m2, or evident proteinuria/haematuria, persisting for more than 3 months. In this study, the eGFR was according to Abbreviated Modification of Diet in Renal Disease Formula (aMDRD):  eGFR = 186 × Creatinine−1.154 × Age−0.203 (× 1 if male, × 0.742 if female). In Taiwan Air Force, only medical record with relative healthy subjects could be involved. The airline service experience is defined as (age-26). But detailed total flight time, severity of gravity for bodies, and back ground diseases may have the impact on the prognosis for kidney diseases. In Taiwan Air Force, only medical record with relative healthy subjects could be involved. The airline service experience is defined as (age-26). However, total flight time, severity of gravity for bodies, and back ground diseases may have the impact on the prognosis for kidney diseases. We made this correction and added this point in the section of Method and limitation. Please see page 3, lines 111-119 and page 29, lines 925-928.

4.5. Study Limitations

  1. This study used the data collected by the Retrospective Generational Tracking Research Plan of the Physical Examination Indicators of Taiwanese Military Aircrew. Since no data on serum cystatin C (Cys C), and eating (living) habits were collected, no analysis of those factors was performed.
  2. Due to the special operating environment and extremely strict screening of Air Force aircrew, only healthy aircrew can continue to serve. Therefore, no stage 3B-5 CKD cases were included, and therefore, it is necessary to be more cautious when inferring the research results.
  3. This study cannot confirm the renal parenchymal damage, such as urine protein, urine RBC positivity, and urine pH >8, since subjects had early-stage CKD and cannot confirm whether the other chronic diseases such as chronic hepatitis B or C infection associated kidney diseases [25].
  4. Most of the subjects in this study are located in the north and south regions of Taiwan (accounting for 83%), which increases the difficulty of finding certain factors, and thus, the generalizability to other regions should be limited.
  5. In Taiwan Air Force, only medical record with relative healthy subjects could be involved. The airline service experience is defined as (age-26). But detailed total flight time, severity of gravity for bodies, and back ground diseases may have the impact on the prognosis for kidney diseases.

  1. The prevalence in a healthy general population for CKD should be very low.  Why is the number still detectable?  How long were these individuals in the air force and what is the change in the study variables from baseline physical examination?

Author Reply: Thank you for your valuable comments. To avoid overdiagnosis, CKD is defined as the presence of kidney damage based on the estimated glomerular filtration rate (eGFR) less than 60 ml/min/1.73m2, or evident proteinuria/haematuria, persisting for more than 3 months in this study. The eGFR was according to Abbreviated Modification of Diet in Renal Disease Formula (aMDRD):  eGFR = 186 × Creatinine−1.154 × Age−0.203 (× 1 if male, × 0.742 if female). We made the comparisons from baseline for 4 years. The results show that the prevalence of CKD was 3.8%, 9.4%, 9.0%, 9.4% in each of the four years (Table 1). Please see page 10, lines 496-513 and page 11 (Table 1).

The comparison of the CKD correlation in previous physical examinations was shown in Table 1. The mean GFR of 93.14±16.01 ml/min/1.73 m2 in the fourth physical examination was the lowest, and that of 107.29±19.34 ml/min/1.73 m2 in the first physical examination was the highest. The prevalence of CKD was 3.8% (8) for the first physical examination, 9.4% (20) for the second physical examination, 9.0% (19) for the third physical examination, and 9.4% (20) for the fourth physical examination. In terms of the CKD stage, in the first physical examination, eight were in stage 1; in the second physical examination, 14 were in stage 1, and six were in stage 2; in the third physical examination, 11 were in stage 1, seven were in stage 2, and one was in stage 3; in the fourth physical examination, 10 were in stage 1, seven were in stage 2, and three were in stage 3. The results showed that the mean GFR was significantly different (p<0.001) in all previous physical examinations, and the Scheffé test (a type of post-hoc, statistical analysis test) showed that the mean GFR in the first physical examination was greater than the means found in the third and fourth physical examinations. Moreover, the mean GFR in the second physical examination was greater than that in the fourth physical examinations, and the mean GFR decreased along the time sequence of physical examinations (p<0.001). The mean GFR was significantly different between the CKD and non-CKD groups in all previous physical examinations (p<0.001).

An English writer should review the paper

Author Reply: Thank you for your valuable comments. We do our best to improve the English errors and sent this revised manuscript to AJE for English editing again.

Last, we are deeply honored by the time and effort you spent reviewing this manuscript. In reviewing and revising our manuscript, we are motivated to read more and thus learn more from your criticisms.

Reviewer 3 Report

Comments and Suggestions for Authors

In the present study titled “Biomedical Evaluation of Early Chronic Kidney Disease in Air Force: Building a Predictive Model from Taiwan Military Healthy Service” authors identified and addressed the increasing prevalence of chronic kidney disease (CKD) among Air Force aircrews in Taiwan, emphasizing the importance of early diagnosis and prevention. It also provides valuable insights into specific risk factors using rigorous statistical analyses, contributing to tailored preventive measures for this occupational group. The focus on a specific population in Taiwan may limit broader applicability, and the lack of an interventional component hinders a comprehensive understanding of effective preventive measures for CKD in aircrews. The manuscript is well written and organized properly.

Author Response

Response to Reviewer 3

[General Comment]

In the present study titled “Biomedical Evaluation of Early Chronic Kidney Disease in Air Force: Building a Predictive Model from Taiwan Military Healthy Service” authors identified and addressed the increasing prevalence of chronic kidney disease (CKD) among Air Force aircrews in Taiwan, emphasizing the importance of early diagnosis and prevention. It also provides valuable insights into specific risk factors using rigorous statistical analyses, contributing to tailored preventive measures for this occupational group. The focus on a specific population in Taiwan may limit broader applicability, and the lack of an interventional component hinders a comprehensive understanding of effective preventive measures for CKD in aircrews. The manuscript is well written and organized properly.

Author Reply: We sincerely appreciate your time and effort spent reviewing this manuscript. We have revised the manuscript thoroughly according to all reviewers’ suggestions and comments. We also reorganized the whole text body (from total pages 48 to 31).  

Last, we are deeply honored by the time and effort you spent reviewing this manuscript. In reviewing and revising our manuscript, we are motivated to read more and thus learn more from your criticisms.

Round 2

Reviewer 1 Report

Comments and Suggestions for Authors

No additional comments.

Thanks.

Comments on the Quality of English Language

NA